# Cell-penetrating protein-recognizing polymeric nanoparticles through dynamic covalent chemistry and double imprinting

Avijit Ghosh [1], Mansi Sharma[1] & Yan Zhao [1] ✉

Molecular recognition of proteins is key to their biological functions and processes such as protein–protein interactions (PPIs). The large binding interface involved and an often relatively flat binding surface make the development of selective protein-binding materials extremely challenging. A general method is reported in this work to construct protein-binding polymeric nanoparticles from cross-linked surfactant micelles. Preparation involves first dynamic covalent chemistry that encodes signature surface lysines on a protein template. A double molecular imprinting procedure fixes the binding groups on the nanoparticle for these lysine groups, meanwhile creating a binding interface complementary to the protein in size, shape, and distribution of acidic groups on the surface. These water-soluble nanoparticles possess excellent specificities for target proteins and sufficient affinities to inhibit natural PPIs such as those between cytochrome c (Cytc) and cytochrome c oxidase (CcO). With the ability to enter cells through a combination of energy-dependent and -independent pathways, they intervene apoptosis by inhibiting the PPI between Cytc and the apoptotic protease activating factor-1 (APAF1). Generality of the preparation and the excellent molecular recognition of the materials have the potential to make them powerful tools to probe protein functions in vitro and in cellulo.

Proteins perform vital tasks, including molecular recognition, catalysis, and transport, which together form the molecular basis for biological functions. Proteins rarely work alone in these processes, but are regulated frequently through binding with other protein partners. Mapping out these protein–protein interactions (PPIs) is key to fundamental biology, as well as drug discovery, because aberrant PPIs are closely associated with many disease developments[1–3]. Chemists are experts at making small molecules to bind proteins at their ligand-binding sites (through drug discovery). However, when it comes to binding proteins on the surface to inhibit PPIs, their efforts are frequently thwarted by the large area needed and an often relatively flat binding interface.

In principle, combined hydrophobic and polar interactions (e.g., hydrogen bonds and electrostatic interactions) are effective for molecular recognition in water, as long as the two binding interfaces are complementary to each other. These interactions are indeed prevalent in both domain- and peptide-mediated PPIs[4]. However, the buried surface area in PPIs easily reaches 1000–2000 $Å^2$, whereas that in the binding of a small molecule in a deep cavity of an enzyme is merely 300–500 $Å^2$[5]. Generally speaking, preorganizing a large number of charged, hydrogen-bonding, and hydrophobic groups over an extended surface area on a synthetic scaffold is an extremely daunting task[6].

Nanoparticles (NPs) represent a potential material to bind proteins, given their comparable surface size. Shea and co-workers created polymeric NP libraries from a cocktail of functional monomers (FMs) and, through careful optimization, turned them into protein binders[7–9]. An alternative method is molecular imprinting, using either

[1]Department of Chemistry, Iowa State University, Ames, IA 50011-3111, USA. ✉e-mail: zhaoy@iastate.edu

a peptide epitope[10–18] or the whole protein[19–22] as the template. Great efforts have been devoted to enhance the protein-binding properties of molecularly imprinted polymers (MIPs)[23–26], though solid-phase synthesis using surface-anchored templates[27,28], orientated immobilization of templates via boronic acids[12], aptamers[29], and other techniques[30].

A peptide epitope as the template has the advantage of avoiding easily denatured biomacromolecules in the imprinting process and is compatible with many imprinting techniques. As a result, epitope-imprinted materials have found many uses in biology[10–18]. Nonetheless, the commonly used C- or N-terminal epitopes may not always be the best choice in practical applications because they are often involved in posttranslational modifications (PTMs). For discontinuous epitopes consisting of separate residues brought together by the folding of the peptide chain, it is much harder or even impossible to find a small molecule analog that can be used as the template. Imprinting of a whole protein, on the other hand, uses the natural protein directly as the template and can yield a binder with a more extensive buried interface. No structural information is needed for the imprinting as long as the protein is available. The difficulty, nonetheless, remains in the execution, given the fragile nature of the large biomolecule. The very few reported methods for whole protein imprinting all have major shortcomings[23]. As it stands, the unsolved problem in protein recognition by NPs remains in the creation of a complementary binding interface to an arbitrary protein in size, shape, and functionality.

In this work, we report a general method to prepare protein-binding nanoparticles via direct imprinting of whole protein. Our method takes advantage of reversible covalent interactions between trifluoromethyl ketone-based FMs and surface lysines on a protein. Dynamic covalent chemistry[31–34] is a powerful method to optimize binding interactions for their targeted molecular guests[35–39]. It enables the FM-containing surface-cross-linked micelles (SCMs), in our case, to equilibrate to the best protein-binding state and a double imprinting procedure fixes this state, affording core-shell NPs with excellent abilities to distinguish proteins by their signature surface lysines, the electrostatic and hydrophobic microenvironments near the binding lysines, and other properties including the surface topology and distribution of acidic groups on the surface of the protein. The generality of the method is demonstrated through proteins with different molecular weights and isoelectric points (PIs). These NPs are shown to inhibit the PPIs between cytochrome c (Cytc) and cytochrome c oxidase (CcO) in vitro. In addition, they enter cells through a combination of energy-dependent and -independent pathways and readily reach the cytosol to inhibit Cytc-triggered apoptosis.

## Results and discussion

For molecular imprinting to work effectively, the FMs need to interact with the template either in a bulk mixture or on the surface of a support, while copolymerization of the FMs with suitable cross-linkers traps the template in the complex form. This trapping, essentially the molecular imprinting process, is imperfect because the FM–template complex, if noncovalent in nature, generally forms and breaks on a time scale much faster than the rate of polymerization. Molecular imprinting of protein is particularly challenging, given the difficulty in the design of FMs that can bind protein surface groups strongly and selectively in water. For a macromolecular template, one also needs to keep it near the surface of the final imprinted material so that it will not be permanently trapped in the polymer network. A related constraint is in the cross-linking density: whereas a high density helps fix the binding groups and polymer network into the proper configurations to recognize the template, it tends to increase the probability of permanent entrapment of the guest. A low cross-linking density, on the other hand, lowers the integrity of the imprinted binding sites and is detrimental to the binding.

Figure 1 illustrates our methods to overcome these difficulties, using dynamic covalent chemistry to optimize the protein–FM complex and a double imprinting strategy to fix the binding groups while constructing the protein-binding interface. Molecule 1a is the protein-binding FM used in our molecular imprinting. Its trifluoromethyl ketone group, being highly electrophilic, forms hemiaminal quickly and reversibly with primary amines[40,41]. An ortho amide group facilitates the formation of hemiaminal 2a by an intramolecular hydrogen bond. Two such units have been reported to bind α amino acids with a large association constant of $K_a \approx 10^7\,M^{-1}$ in acetonitrile[42].

FM 1a has a low water solubility but can be solubilized in water by the micelles of 3. Surfactant 3 has a tripropargylammonium head group and a polymerizable methacrylate at the end of its hydrophobic tail. Micelles formed by this surfactant are covered with a layer of terminal alkynes on the surface and are readily cross-linked by diazide 4, via the highly efficient Cu(I)-catalyzed alkyne–azide click reaction. The 1:1 ratio between 3 and 4 leaves behind multiple alkynes on the surface of the resulting SCM for further click functionalization. Part of the reason for using SCMs instead of uncross-linked micelles to interact with the protein is that uncross-linked surfactants can easily denature proteins, whereas SCMs with their hydrophobic tails tucked inside have little surface activity[43].

Our hypothesis is that, once 1a diffuses out of the SCM, it can interact with the reactive surface lysines on the templating protein in the solution. Meanwhile, hydrophobic voids will be left behind inside the SCM. For a dynamic, noncovalently stabilized micelle, the surfactants will rearrange to eliminate such hydrophobic voids that otherwise would have to be filled with water molecules. For a cross-linked micelle, the rearrangement is more difficult, creating a hydrophobic driving force for the FMs to reenter. Hemiaminal easily hydrolyzes in aqueous solution[40,41], and the 1a–protein complex is not expected to be stable in water. However, it is thermodynamically favorable for the FMs on the 1a–protein complex to reenter the SCM, to fill the hydrophobic voids while being protected from hydrolysis. From the perspective of the protein template, after reacting with the FMs, it becomes equipped with several hydrophobic anchors or tentacles to bind to the SCM, ready for imprinting.

Our next step is to capture the equilibrated SCM–protein complex through polymerization/cross-linking. This is essentially a combined covalent/noncovalent molecular imprinting which creates a polymeric NP having an imprinted site with the distribution of trifluoromethyl ketone moieties matching signature lysines on the protein, plus any electrostatic and hydrophobic interactions captured during the polymerization/cross-linking.

The SCM contains DMPA (2,2-dimethoxy-2-phenylacetophenone, an oil-soluble free radical photoinitiator), with water-soluble MBAm (N,N′-methylenebisacrylamide) in the aqueous solution. Upon UV irradiation, the methacrylamide group of 1a copolymerizes with the methacrylate of 3 in the SCM, to fix the FM and SCM in the protein-binding configuration. During this imprinting process, the DMPA photoinitiator initiates free radical polymerization within the SCM due to its hydrophobicity, while water-soluble MBAm molecules are present in the aqueous solution. Since the growing polymer chain is confined inside the SCM, it polymerizes only those MBAm molecules diffused to the surface of the micelle. Polymerization/cross-linking then installs a layer of hydrogen-bonding amide groups on the surface of NP_A, some of which are fixed in the protein-binding positions. Previously, MBAm has been found to enhance the imprinting and binding of amphiphilic guests such as 4-nitrophenyl-α-D-glucopyranoside by 180-fold[44].

The surface- and the core-cross-linking of the micelles were monitored by [1]H NMR spectroscopy (Supplementary Fig. 6), showing characteristic changes (e.g., disappearance of polymerizable vinyl protons). SCM was confirmed not to cause any significant conformational changes in the templating protein by CD spectroscopy

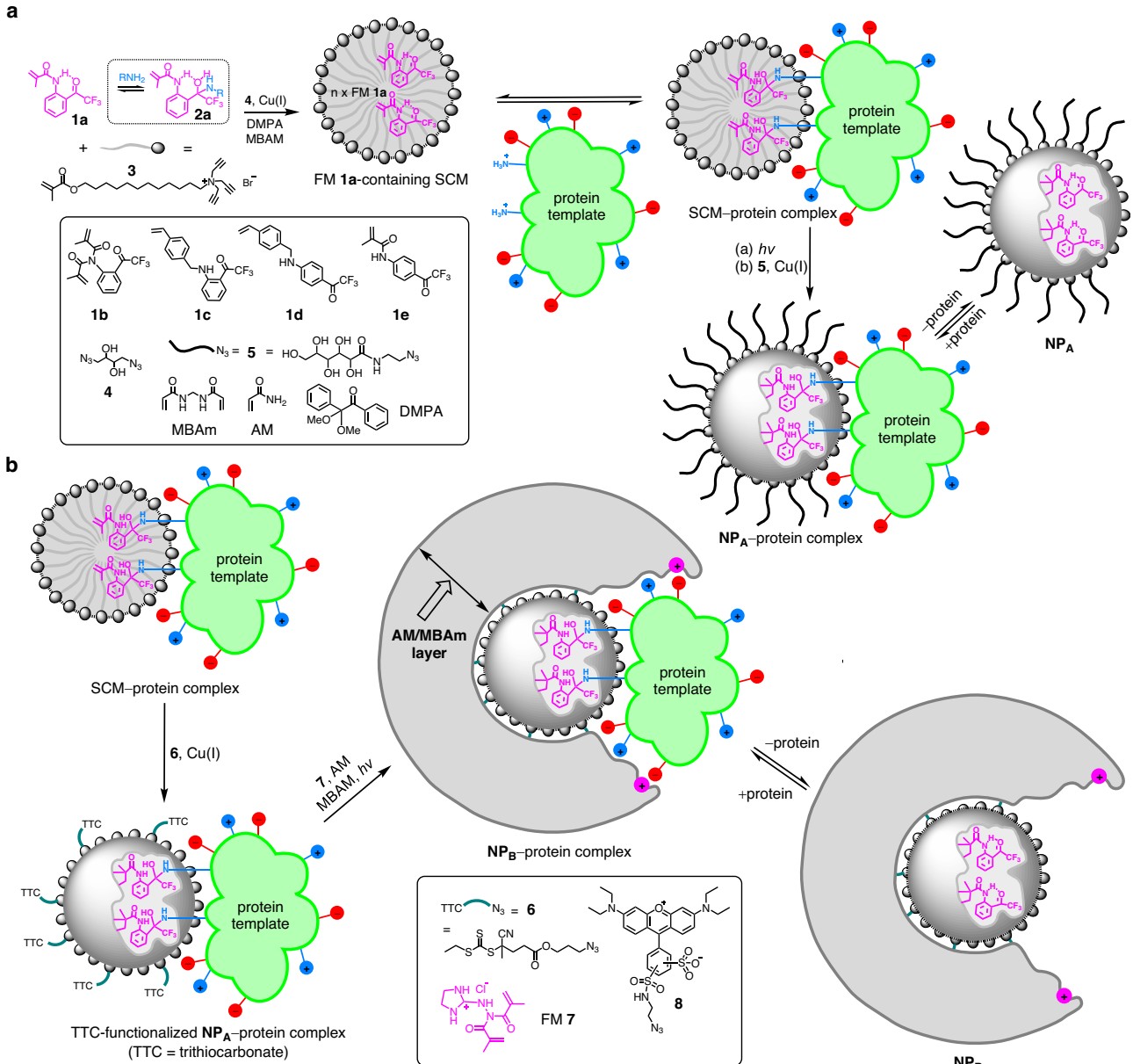

**Fig. 1 | Reaction schemes for the preparation of protein-binding nanoparticles.**
**a** Preparation of NP$_A$ using the FM 1a-containing surface-cross-linked micelle (SCM).
**b** Preparation of NP$_B$ via a double imprinting strategy. The red and blue spheres on the protein represent positively and negatively charged surface groups. The magenta spheres on NP$_B$ represent carboxylate-binding functional groups installed through polymerization of FM 7.

(Supplementary Fig. 7). Removal of the protein template during purification was confirmed by MALDI-TOF mass spectrometry (Supplementary Fig. 8). The size and zeta potential of the nanoparticles were also determined (Supplementary Figs. 9–13).

It is difficult to predict which lysines on a protein will react with **1a**. Nonetheless, under dynamic covalent chemistry[31–34], the equilibrium will favor a region of protein with several reactive lysines close by, so that multiple hydrophobic anchors can enter the micellar core simultaneously and be stabilized. Other interactions that can influence this equilibrium include any potential hydrophobic contact between the SCM and the protein, as well as any ion pairing and hydrogen-bonding interactions that can be established. The mechanism does not rule out any complexed **1a** left in the aqueous solution, but an exposure of large hydrophobic groups to water is unfavorable, and hemiaminal is unstable in water.

In Fig. 1a, the cross-linked micelle is generally functionalized with surface ligand **5** in the last step for facile purification of the NP

products. Without surface functionalization, the terminal alkynes can be used for the installation of a range of other surface ligands (such as fluorescent labels described later in the paper). To create a more extended binding interface, we clicked trithiocarbonate- or TTC-functionalized azide **6** onto the surface of **NP$_A$** (Fig. 1b), for photo RAFT polymerization[45]. These surface-anchored initiators allow copolymerization between water-soluble acrylamide (AM) and MBAm to take place on the surface of **NP$_A$**, to create a layer of hydrophilic polymer around the bound protein. The AM/MBAm outer layer should further enhance the binding affinity and selectivity of the doubly imprinted **NP$_B$**. Not only so, a guanidinium-containing FM **7** can be included in the photo RAFT polymerization. After polymerization, these FMs will turn into carboxylate-binding functionalities (shown as magenta-colored spheres with a positive sign in Fig. 1b), complementary to some of the acidic groups on the surface of the protein.

Support for the trifluoromethyl ketone-mediated imprinting of protein comes from binding studies using isothermal titration

**Table 1 | Binding properties of NP$_A$(lysozyme) determined by isothermal titration calorimetry (ITC).[a]**

| Entry | Host | Guest | FM | $K_a$ (× 10⁴ M⁻¹) | $K_{rel}$ | −ΔG (kcal/mol) | −ΔH (kcal/mol) | TΔS (kcal/mol) |
|---|---|---|---|---|---|---|---|---|
| 1 | **NP$_A$**(lysozyme) | lysozyme | **1a** | 57.3 ± 2.8 | 1.00 | 7.86 | 24.45 ± 0.39 | −16.59 |
| 2 | **NP$_A$**(lysozyme) | lysozyme | **1b** | 47.8 ± 3.4 | 0.83 | 7.75 | 17.60 ± 0.48 | −9.85 |
| 3 | **NP$_A$**(lysozyme) | lysozyme | **1c** | 33.0 ± 1.7 | 0.58 | 7.53 | 15.60 ± 0.30 | −8.07 |
| 4 | **NP$_A$**(lysozyme) | lysozyme | **1d** | 29.9 ± 1.4 | 0.52 | 7.47 | 13.74 ± 0.21 | −6.27 |
| 5 | **NP$_A$**(lysozyme) | lysozyme | **1e** | 32.5 ± 1.6 | 0.57 | 7.52 | 15.11 ± 0.28 | −7.59 |
| 6 | NINP | lysozyme | **1a** | <0.12 [b] | <0.002 | - | - | - |
| 7 | **NP$_A$**(lysozyme) | lysozyme | none | 0.92 ± 0.4 | 0.016 | 5.41 | 2.34 ± 0.75 | 3.07 |
| 8 | **NP$_A$**(lysozyme) | BSA | **1a** | 2.67 ± 0.7 | 0.047 | 6.04 | 2.23 ± 0.89 | 3.81 |
| 9 | **NP$_A$**(lysozyme) | HRP | **1a** | <0.004 [b] | <0.00007 | - | - | - |
| 10 | **NP$_A$**(lysozyme) | α-amylase | **1a** | <0.001[b] | <0.00002 | - | - | - |
| 11 | **NP$_A$**(lysozyme) | Cytc | **1a** | <0.06 [b] | <0.001 | - | - | - |
| 12 | **NP$_A$**(lysozyme) | chymotrypsin | **1a** | <0.05 [b] | <0.001 | - | - | - |
| 13 | **NP$_A$**(lysozyme) | OVA | **1a** | 3.24 ± 0.8 | 0.057 | 6.15 | 3.15 ± 0.96 | 3.00 |
| 14 | **NP$_A$**(lysozyme) | transferrin | **1a** | <0.04 [b] | <0.0007 | - | - | - |
| 15 | **NP$_A$**(lysozyme) | trypsin | **1a** | <0.05 [b] | <0.0009 | - | - | - |

[a]**NP$_A$**(lysozyme) was prepared using an optimized ratio of [**3**]/[**4**]/[**5**]/[MBAm]/[FM]/[lysozyme] = 50:50:100:100:8:1. Binding constants were determined by triplicate, independent ITC titrations at 298 K in 10 mM HEPES buffer (pH = 7.5), with the relative errors among the runs in the range of 4–7% (typically <5%). The errors shown in the Table are from curve-fitting for representative titrations, with the corresponding titration curves given in the Supplementary Information (Supplementary Fig. 14).
[b]The binding constant could not be determined accurately by ITC due to the weak binding. Because of the weak binding, the titration curve fits poorly and has large uncertainties in the estimated binding constant.

calorimetry (ITC). As shown in Table 1, **NP$_A$**(lysozyme), i.e., the imprinted polymeric nanoparticle obtained using lysozyme as the template, displays a binding constant of $K_a = (57.3 ± 2.8) × 10^4$ M⁻¹ for lysozyme in aqueous buffer, equivalent to a binding free energy of $−ΔG = 7.86$ kcal/mol. Meanwhile, the nonimprinted nanoparticle (NINP) prepared without the protein template shows minimal binding, consistent with successful imprinting (entry 6). The functional monomer is critical to the imprinting, as **NP$_A$**(lysozyme) prepared without **1a** exhibits a much weaker binding (entry 7).

**NP$_A$**(lysozyme) was prepared with a formulation of [**3**]/[**4**]/[**5**]/[MBAm]/[**1a**]/[lysozyme] = 50:50:100:100:8:1. We typically choose a surfactant/template of ratio of 50:1 in the preparation because each cross-linked SCM contains approximately 50 surfactants according to dynamic light scattering (DLS)[46]. Each surfactant has three propargyl groups and each azide cross-linker 2 azides; a 1:1 ratio of [**3**]/[**4**] means one equivalent of surface alkynes will be left for further click functionalization if the click cross-linking happens perfectly. The amount of MBAm and **1a** was optimized experimentally for the highest binding constant achievable. The surface ligand (**5**) is generally used in excess so that the final NP with as many of this ligand on the surface can be precipitated from acetone. Lysozyme is known to have 6 reactive lysines[47]. Our screening shows that an 8:1 ratio of FM/template affords **NP$_A$**(lysozyme) with the strongest binding for its templating protein.

We examined a number of trifluoromethyl ketone FMs **1a**–**1e** (Fig. 1). Even though all these FMs work for the imprinting, the ortho-substituted ones generally work better than the para-substituted and having a secondary amide group is particularly beneficial (Table 1, entries 1–5), consistent with the intramolecular hydrogen bond that can activate the ketone for the nucleophilic attack of amine and/or stabilize the product[40,41]. Tertiary amide derivative **1b** also gives good results, possibly because of its higher hydrophobicity and/or double polymerizable vinyl groups.

To further support the role of the trifluoromethyl ketone in the binding, we performed inhibition experiments and titrated **NP$_A$**(lysozyme) with lysozyme in the presence of 0, 1, 2.5, and 5 equivalents of 3-amino-1-propanol in the solution. The amine additive is expected to compete with the protein for the trifluoromethyl ketone binding groups. Indeed, the binding between the protein and **NP$_A$**(lysozyme) decreased from $57.3 × 10^4$ M⁻¹ (Table 1, entry 1) quickly

to $5.0 × 10^4$ M⁻¹ (Supplementary Fig. 17a), to $2.4 × 10^4$ M⁻¹ (Supplementary Fig. 17b), and then to $<3 × 10^3$ M⁻¹ (Supplementary Fig. 17c), respectively, indicating that amines are critical to the binding.

**NP$_A$**(lysozyme) is highly selective for its protein template (Table 1, entries 8–15). The same method works well for bovine serum albumin (BSA), horse radish peroxidase (HRP), amylase, and Cytc, supporting the generality of the method (Fig. 2a). Cross-reactivities are generally low, as the nontemplating proteins display much weaker bindings. When the lysozyme in the presence of 2 equivalents of another protein (BSA, HRP, α-amylase, Cytc, chymotrypsin, OVA, transferrin, or trypsin) is titrated into **NP$_A$**(lysozyme), the binding constants obtained average $K_a = (53.7 ± 8.4) × 10^4$ M⁻¹ (Supplementary Table 2 and Supplementary Fig. 16), experimentally the same as that without the competitive protein, i.e., $K_a = (57.3 ± 2.8) × 10^4$ M⁻¹. (Table 1, entry 1). This result further underscores the selectivity of the materials.

As discussed earlier, our hypothesis is that dynamic covalent chemistry will favor a region of protein with multiple reactive lysines close by, among other factors. Since a spherical SCM can only accommodate and stabilize a limited number of protein-conjugated FMs due to its geometrical constraint, we expect the binding affinity of **NP$_A$** would not differ greatly if these FMs are the main contributors to the binding (as supported by the inhibition experiments discussed above) and a similar number of the hemiaminal bonds are formed in all protein–NP pairs. Even if a protein contains many reactive lysines, only those with their hemiaminal of **1a** inserted into the SCM, stabilized (through water exclusion), and captured covalently (through polymerization) will contribute to the protein binding at the end.

The above postulation was confirmed experimentally. The number of reactive lysines on our protein templates varies from 6 to 32[47–51]. If the N-terminal amine is counted, it will add 1 to the above number. Yet, the binding constants of the different NPs toward their templating proteins were $(54 ± 4) × 10^4$ M⁻¹, across different proteins and their NP binders (Fig. 2a).

On the other hand, as the number of reactive lysines on a protein template increases, a larger FM/template ratio is needed in the NP preparation. A small amount of FM will be unable to shift the equilibrium to the state with multiple adjacent lysines anchored into the SCM by the hemiaminals formed. This was also confirmed experimentally. As shown in Fig. 2b, the optimal FM/template ratio in the **NP$_A$**

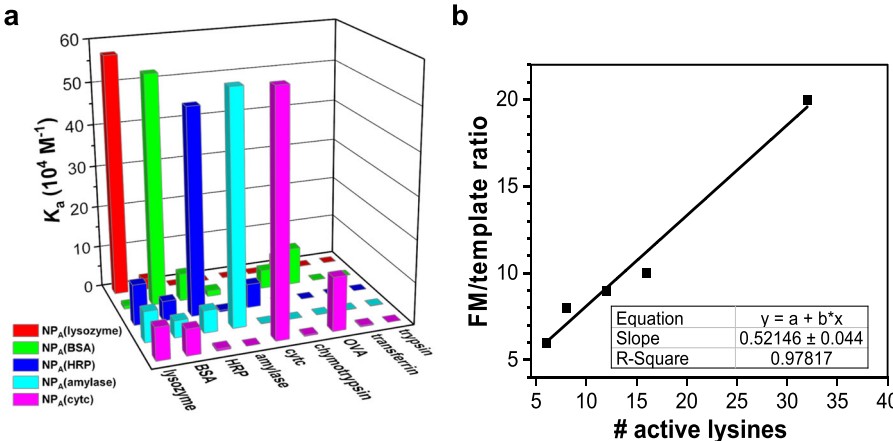

**Fig. 2 | Binding properties of NP_A. a** Binding constants of five different NP_A hosts for different proteins, demonstrating the selectivity in the binding. **b** Relationship between the number of active lysine residues on a protein and the optimal FM 1a/template ratio used in NP_A preparation. The binding constants for the nontemplating proteins could not be determined accurately by ITC due to the weak binding. Because of the weak binding, the titration curves fit poorly and have large uncertainties in the estimated binding constants.

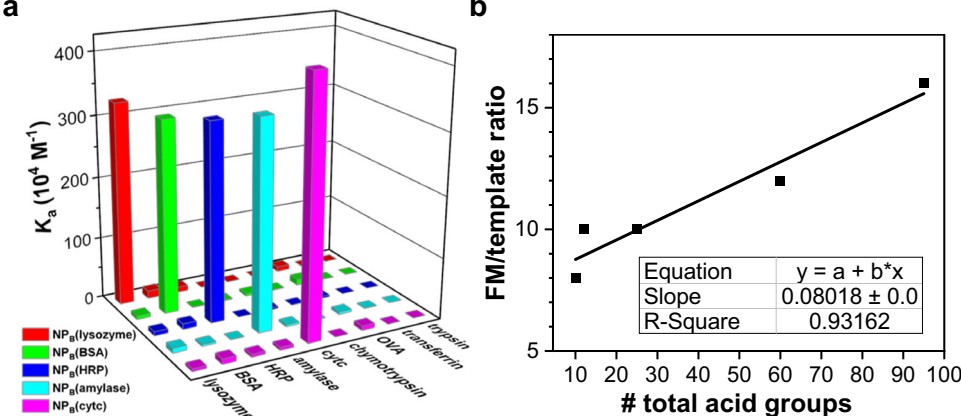

**Fig. 3 | Binding properties of NP_B. a** Binding constants of five different NP_B hosts for different proteins, demonstrating the selectivity in the binding; **b** Relationship between the number of active acidic residues on a protein and the optimal FM 7/template ratio used in NP_B preparation. The binding constants for the nontemplating proteins could not be determined accurately by ITC due to the weak binding. Because of the weak binding, the titration curves fit poorly and have large uncertainties in the estimated binding constants.

preparation correlates linearly with the number of reactive lysines on the protein templates.

The results so far are encouraging because they confirm that dynamic covalent chemistry and molecular imprinting in combination work well as a general method for making protein-binding NPs. Although the micromolar binding affinities may be sufficient for some applications, we would like to improve the binding strength and selectivity further.

To our delight, the outer AM/MBAm layer (Fig. 1b) further strengthens the binding (Fig. 3a). The optimized procedure employs a 3:1 w/w ratio of AM/MBAm. As shown in Table 2, this outer layer enables the binding of **NP_B**(lysozyme) to reach $K_a = (11.8 \pm 2.0) \times 10^5\,\mathrm{M}^{-1}$ and increase further with FM **7** to $32.8 \times 10^5\,\mathrm{M}^{-1}$. This value is nearly six times higher than the binding constant of **NP_A**(lysozyme) for the same protein. Without TTC-surface functionalization of **NP_A**, UV irradiation of a mixture of **NP_A**(lysozyme) and AM/MBAm did not enhance the binding. This supports the importance of the TTC-surface functionalization (and the subsequent RAFT polymerization)[45]. Consistent with the involvement of the guanidinium–carboxylate salt bridges in the outer layer, the optimal FM **7**/template ratios exhibit a positive correlation with the total number of acidic groups in the protein templates (Fig. 3b).

Our data indicate that the outer AM/MBAm layer on **NP_B** is a minor contributor to the binding in comparison to the covalent hemiaminal bonds. **NP_A**(lysozyme), for example, binds its protein template with a free energy of $-\Delta G = 7.86\,\mathrm{kcal/mol}$ (Table 1, entry 1). Adding the AM/MBAm layer (without the FM **7**) only increases the binding energy by merely 0.41 kcal/mol. Even with the FM **7** in the AM/MBAm layer, the binding energy increases by 1.0 kcal/mol. Therefore, the major contributor to the protein binding is the reversible hemiaminal covalent bond and all the other interactions, whether in **NP_A** or **NP_B**, are secondary. Since the AM/MBAm layer only strengthens the protein binding by no more than 1 kcal/mol, molecular imprinting in this outer layer is moderate with the current method, especially considering the additional binding interface potentially formed (in the outer layer).

Table 3 shows that **NP_A** has a hydrodynamic diameter of about 5 nm and the AM/MBAm layer adds 5–6 nm to the diameter. During our optimization for the AM/MBAm outer shell polymerization, we were initially concerned that too thick a polymer layer would bury the

**Table 2 | Binding properties of $NP_B$(lysozyme) determined by isothermal titration calorimetry (ITC).[a]**

| Entry | Host | Guest | Template/[7] | $K_a$ (× $10^5$ $M^{-1}$) | $K_{rel}$ | $-\Delta G$ (kcal/mol) | $-\Delta H$ (kcal/mol) | $T\Delta S$ (kcal/mol) |
|---|---|---|---|---|---|---|---|---|
| 1 | $NP_B$(lysozyme) | lysozyme | 1:0 | 11.8 ± 2.0 | 0.36 | 8.28 | 23.36 ± 0.80 | −15.08 |
| 2 | $NP_B$(lysozyme) | lysozyme | 1:1 | 15.6 ± 3.5 | 0.48 | 8.45 | 19.44 ± 1.19 | −10.99 |
| 3 | $NP_B$(lysozyme) | lysozyme | 1:2 | 20.7 ± 4.6 | 0.63 | 8.62 | 36.48 ± 1.68 | −27.86 |
| 4 | $NP_B$(lysozyme) | lysozyme | 1:4 | 22.2 ± 2.4 | 0.68 | 8.66 | 24.82 ± 0.61 | −16.16 |
| 5 | $NP_B$(lysozyme) | lysozyme | 1:6 | 27.1 ± 5.6 | 0.83 | 8.78 | 21.84 ± 1.07 | −13.06 |
| 6 | $NP_B$(lysozyme) | lysozyme | 1:8 | 32.8 ± 7.6 | 1 | 8.89 | 31.17 ± 1.56 | −22.28 |
| 7 | $NP_B$(lysozyme) | lysozyme | 1:10 | 30.6 ± 4.2 | 0.93 | 8.85 | 55.39 ± 1.64 | -46.54 |
| 8 | NINP | lysozyme | 0:0 | <0.047 [c] | <0.0014 | - | - | - |
| 9 | NINP | lysozyme | 0:8 | <0.015 [c] | <0.0004 | - | - | - |
| 10 | $NP_B$(lysozyme)[b] | lysozyme | 1:8 | 0.59 ± 0.1 | 0.0180 | 6.51 | 10.73 ± 4.16 | −4.22 |
| 11 | $NP_B$(lysozyme) | BSA | 1:8 | 1.24 ± 0.3 | 0.0378 | 6.95 | 9.81 ± 2.17 | −2.85 |
| 12 | $NP_B$(lysozyme) | HRP | 1:8 | 0.86 ± 0.3 | 0.0262 | 6.73 | 11.28 ± 8.58 | −4.55 |
| 13 | $NP_B$(lysozyme) | α-amylase | 1:8 | <0.004 [c] | <0.0001 | - | - | - |
| 14 | $NP_B$(lysozyme) | Cytc | 1:8 | <0.004 [c] | <0.0001 | - | - | - |
| 15 | $NP_B$(lysozyme) | chymotrypsin | 1:8 | <0.0001 [c] | <0.000003 | - | - | - |
| 16 | $NP_B$(lysozyme) | OVA | 1:8 | 0.82 ± 0.1 | 0.025 | 6.70 | 5.13 ± 1.34 | 1.58 |
| 17 | $NP_B$(lysozyme) | transferrin | 1:8 | <0.08 [c] | <0.002 | - | - | - |
| 18 | $NP_B$(lysozyme) | trypsin | 1:8 | <0.01[c] | <0.0003 | - | - | - |

[a]$NP_B$(lysozyme) was prepared in the following formulation unless otherwise indicated: [3]/[4]/[5]/[MBAm]/[FM]/[lysozyme] = 50:50:100:100:8:1 for $NP_A$; [6]/[7]/[MBAm]/[AM]/[3] = 0.05:0.2:4.8:31.6: 1 for $NP_B$(lysozyme). Binding constants were determined by triplicate, independent ITC titrations at 298 K in 10 mM HEPES buffer (pH = 7.5), with the relative errors among the runs in the range of 4–7% (typically <5%). The errors shown in the Table are from curve-fitting for representative titrations, with the corresponding titration curves given in the Supplementary Information (Supplementary Fig. 15).
[b]$NP_A$(lysozyme) was prepared without FM **1a** but FM **7** was used in the outer layer preparation for $NP_B$(lysozyme).
[c]The binding constant could not be determined accurately by ITC due to the weak binding. Because of the weak binding, the titration curve fits poorly and has large uncertainties in the estimated binding constant.

**Table 3 | Size and zeta potential of $NP_A$ and $NP_B$.[a]**

| Entry | NP | DLS diameter (nm) | Zeta potential (mV) |
|---|---|---|---|
| 1 | $NP_A$(lysozyme) | 5.0 ± 0.4 | 26 ± 7 |
| 2 | $NP_A$(BSA) | 4.6 ± 0.2 | 25 ± 8 |
| 3 | $NP_A$(HRP) | 4.6 ± 0.2 | 28 ± 9 |
| 4 | $NP_A$(amylase) | 4.7 ± 0.2 | 30 ± 10 |
| 5 | $NP_A$(Cytc) | 4.7 ± 0.3 | 30 ± 7 |
| 6 | $NP_B$(lysozyme) | 11.6 ± 0.2 | 37 ± 13 |
| 7 | $NP_B$(BSA) | 9.5 ± 0.3 | 40 ± 11 |
| 8 | $NP_B$(HRP) | 10.8 ± 0.7 | 41 ± 16 |
| 9 | $NP_B$(amylase) | 10.3 ± 0.5 | 40 ± 10 |
| 10 | $NP_B$(Cytc) | 11.5 ± 0.4 | 39 ± 11 |

[a]Hydrodynamic diameters and zeta potential of nanoparticles were determined in water at 298 K. Data were presented as the mean ± SE, n = 3 independent experiments.

protein template and permanently trap it. Nonetheless, we found that the size of $NP_B$ plateaued to about 10–11 nm even if large amounts of AM/MBAm were used in the second polymerization. As mentioned above, the TTC-surface functionalization of $NP_A$ was key to the formation of the outer polymer layer, consistent with the photo RAFT polymerization being initiated off the surface of $NP_A$. Photo RAFT is a slow process and takes about 30 h to complete[45]. During this process, the AM and MBAm molecules have to diffuse into the polymer layer to encounter photo-generated radicals for polymerization. Once a certain thickness is reached, it is likely that such diffusion into the cross-linked polymer layer gets increasingly difficult, preventing further growth of the polymer layer.

It should be noted that the protein templates studied include both highly acidic ones (amylase with PI ≈ 3.5 and BSA with PI ≈ 5) and basic ones (lysozyme and Cytc with PI ≈ 11), with M.W. ranging from 12 to 80 KDa (Supplementary Table 1). Yet, Figs. 2a, 3a indicate that all of them could interact selectively with their corresponding nanoparticle hosts. The zeta potentials of our protein-binding nanoparticles prepared range from 25–41 mV (Table 3). Thus, generic electrostatic interactions do not play a large role in the binding. These cross-linked micelles are rigid polymeric nanoparticles with extensive cross-linking. The negative charges on a negatively charged protein are typically distributed over the surface of the protein instead of being concentrated in one region. The generic electrostatic interactions between the protein and our nanoparticles apparently are unable to compete with the lysine–trifluoromethyl ketone covalent interactions.

The above study demonstrates that highly selective protein-binding NPs can be prepared using our method. Among all the protein templates studied, Cytc is the smallest one, with an MW of only 12 KD. Yet, it contains an abundance of lysines (Fig. 4a). As a result, despite its many positive charges (PI ≈ 11), its binding with (its corresponding) $NP_B$ is the strongest, as shown in Fig. 3a.

Cytc in mitochondria is involved in the electron transport chain (ETC), shuttling electrons from the cytochrome $bc_1$ complex ($bc_1$) to cytochrome c oxidase (CcO)[52]. As shown in Fig. 4a, some lysines residues (with red text labels) are involved in the Cytc–CcO PPI and others are away from the binding interface. Which lysines would $NP_B$(Cytc) bind when interacting with Cytc and is the binding strong enough to compete with the PPI?

To answer the question, we employed a widely utilized CcO assay, in which the Cytc oxidation is monitored by the decrease of optical density at 550 nm. Figure 4b shows that the addition of 50 μM NINP has a negligible effect on the oxidation. However, as different amounts of $NP_B$(Cytc) were added to the solution, the oxidation reaction was clearly inhibited, in a concentration-dependent fashion. When the percent inhibition is plotted against the concentration of $NP_B$(Cytc) in the solution, the inhibition shows a saturation behavior. More interestingly, when the curve is fitted to a 1:1 binding isotherm (Fig. 4c),

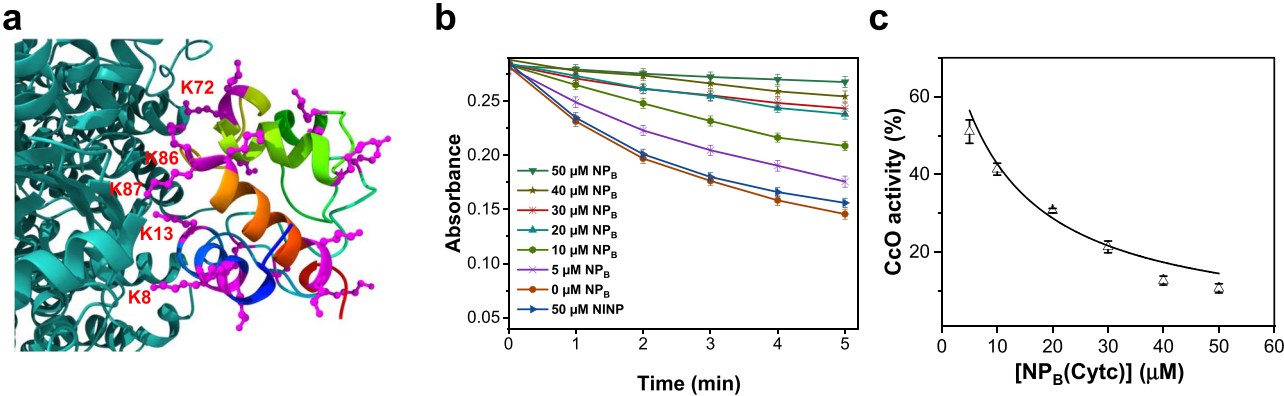

**Fig. 4 | Structure of Cytc−CcO complex and effects of binding of $NP_A$ and $NP_B$ on CcO activity. a** Cytc complex with CcO (PDB ID: 5IY5); the lysines (K) in Cytc are highlighted in magenta, with those involved in the PPIs labeled with red text. The structure shown in teal is CcO. **b** Control CcO assay in comparison to those in the presence of different concentrations of NINP and $NP_B$(Cytc). The experiments were run in triplicates with the indicated errors. **c** CcO activity as a function of the concentration of $NP_B$(Cytc). The smooth theoretical curve is from nonlinear least-squares fitting of the data to a 1:1 binding isotherm. Data were presented as the mean ± SE, $n$ = 3 independent experiments.

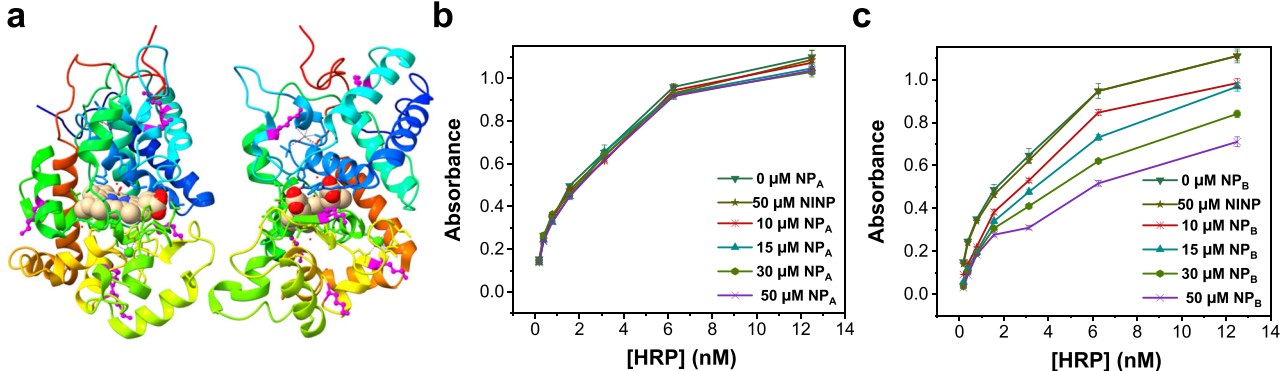

**Fig. 5 | Structure of HRP and effects of binding of $NP_A$/$NP_B$ on HRP activity. a** Crystal structure of HRP (PDB ID: 1w4w) viewed from two directions. The peptide chain is colored from blue (N-terminus) through the rainbow spectrum to red (C-terminus). The lysines (K) are highlighted in magenta. **b** Effects of $NP_A$(HRP) and NINP on HRP activity determined by the TMB assay. **c** Effects of $NP_B$(HRP) and NINP on HRP activity determined by the TMB assay. Data were presented as the mean ± SE, $n$ = 3 independent experiments.

an apparent binding constant of $K_a = (1.0 ± 0.3) × 10^5 M^{-1}$ is obtained. This value is smaller than the actual binding constant between $NP_B$(Cytc) and Cytc determined by ITC, $K_a = 41.4 × 10^5 M^{-1}$. Nonetheless, a correlation between the CcO activity and the $NP_B$(Cytc) concentration suggests that binding was responsible for the inhibition.

As mentioned earlier, our imprinting method relies on the signature surface lysines of a protein, but it is difficult to pinpoint the particular lysines involved, since the dynamic covalent chemistry is a thermodynamic process dependent on multiple factors. With an appropriate functional assay, however, some information can be extracted regarding the lysines involved. In the case of Cytc, since the NP binding clearly inhibits the electron transfer across the PPI interface, the binding cannot be at the opposite end of the PPI. Meanwhile, the much lower apparent binding constant obtained from the inhibition than that from ITC indicates that the inhibitory efficiency is low. From the latter consideration, the NP binding most likely does not occur at the part of Cytc directly involved in the CcO PPI. The fact that $NP_B$(Cytc) can impact the electron transfer suggests that some of the lysines at the CcO-binding interface may be involved in the NP binding or at least the binding with $NP_B$(Cytc) can sterically interfere with the CcO binding, to the point that the electron transfer is negatively impacted.

Functional assays can also help us understand the location of the NP binding sites on some other proteins. HRP has 6 lysines, with most

of them quite a distance away from the heme active site (Fig. 5a). Thus, the NP binding is not anticipated to impact the enzyme's activity as long as the bound NP does not block the entrance of the active site. One of the most widely used HRP assays involves oxidation of 3,3′,5,5′-tetramethylbenzidine (TMB)[53]. The substrate is oxidized by hydrogen peroxide with the help of HRP to produce a blue-colored product that can be monitored by absorption at 655 nm.

As shown in Fig. 5b, as different amounts of HRP are incubated with a mixture of 0.4 mM TMB and 2.0 mM hydrogen peroxide in 200 mM phosphate−citrate buffer (pH 5.0), the absorbance at 655 nm after 4 min increase with higher enzyme concentrations, indicating larger amounts of oxidized products formed. Neither 10−50 μM of $NP_A$(HRP) nor 50 μM of NINP changes the degree of oxidation. Interestingly, when $NP_B$(HRP), the particle with an outer polymer layer, is used in a similar experiment, oxidation of TMB is clearly inhibited by the nanoparticles (Fig. 5c). These results are possible if the imprinted nanoparticles bind the enzyme away from the active site but not on the opposite side. With such an arrangement, the substrate can enter the active site unhinderedly when $NP_A$(HRP) binds to the enzyme. $NP_B$(HRP), on the other hand, has an additional polymer layer 2−3 nm thick, surrounding the bound HRP. If the active site faces sideways from the NP−enzyme binding interface and the polymer layer is sufficiently thick, it can partly or completely block the active site, limiting the access of the substrate.

Cytc is a multifunctional protein. In addition to its role in the ETC, it also triggers apoptosis when released from mitochondria[54,55]. Apoptosis occurs as Cytc binds to the apoptotic protease activating factor-1 (APAF1) and forms an apoptosome that activates caspase-9. The association constant between Cytc and APAF1 is about $K_a = 20.4 \times 10^5\,M^{-1}$[56], about half of the value between Cytc and $NP_B(Cytc)$. Thus, the NP should compete effectively with APAF1 for the Cytc binding and likely intervene in apoptosis if it can be delivered into cell.

NPs can enter cells through a number of different pathways depending on their size, charge, and amphiphilicity[57,58]. Cationic NPs are generally internalized by cells more readily than neutral and anionic ones[59–62], due to their strong interactions with the anionically charged proteoglycans on the surface of most cells[63]. However, they also tend to have higher cytotoxicity[57].

To understand the intracellular delivery of $NP_B(Cytc)$, our strongest receptor for Cytc, we first labeled the nanoparticle with an azido-lissamine derivative **8** using the click chemistry, following similar procedures in Fig. 1b. The labeling was performed on $NP_A(Cytc)$, before TTC azide **6** was clicked. When MDA-MB-231 cells were incubated with the fluorescently labeled $NP_B(Cytc)^*$ for 1 h and washed (to remove external nanoparticles), confocal laser scanning microscopy (CLSM) indicates that the NPs indeed have entered the cells. Prior to imaging, the cells were stained with a nucleus dye (Hoechst 33342), which gives a blue emission under the microscope. The lissamine-labeled NPs give a red emission. Figure 6a–f shows that the NPs stay in the cytosol quite uniformly, outside the nucleus.

Importantly, $NP_B(Cytc)^*$ displayed minimal cytotoxicity in the MTT assay, even at a concentration of 100 μM over a prolonged time (Fig. 6g). It made little difference whether the cells were incubated for 24 or 48 h. The uptake efficiency of the NPs was determined by fluorescence-activated cell sorting (FACS). Figure 6h shows that the % uptake increases with an increase of the NP concentration and reaches nearly 80% at an $NP_B(Cytc)^*$ concentration of 25 μM after 1 h of incubation.

NPs 10–100 nm in size usually are taken up by cells via energy-dependent endocytosis[64]. Small cationic nanoparticles (5–10 nm[65] or even larger[66]), however, can enter cells through direct membrane penetration. To probe the uptake mechanism, we measured the uptake efficiency in the presence of various inhibitory reagents (Fig. 6i)[67]. 2-Deoxy-D-glucose (DOG)/sodium azide depletes ATP in cells and suppresses all energy-dependent processes. Figure 6i shows that the duo indeed inhibits nearly 60% of $NP_B(Cytc)^*$ entry. This result suggests that the nanoparticles enter cells through both direct, energy-independent internalization and energy-dependent pathways. Methyl-β-cyclodextrin (MβCD) removes cholesterol from cell membranes and is known to inhibit CLIC/GEEC endocytosis[58] and lipid raft-mediated uptake, including membrane fusion processes and caveolae-mediated endocytosis[68,69]. The large negative effect of MβCD on the $NP_B(Cytc)^*$ entry indicates that these processes could be the main pathways for the nanoparticles to enter cells or the energy-independent cell entries require cholesterol to be efficient. In contrast, inhibitors for clathrin-mediated endocytosis (CME) including hypertonic sucrose (which traps clathrin in microcages) and chlorpromazine (CPZ, which inhibits AP2 involved in clathrin-mediated endocytosis)[67], have a less significant inhibitory effect on the $NP_B(Cytc)^*$ entry. Lastly, amiloride also shows a small inhibitory effect, suggesting that micropinocytosis is not a major pathway[70].

The above studies indicate that our NPs can enter cells readily and both energy-dependent and -independent pathways are involved. To probe whether $NP_B(Cytc)^*$ upon entering cells can intervene Cytc-triggered apoptosis, we first incubated MDA-MB-231 cells with these nanoparticles for 1 h to allow cellular uptake. The cells were then treated with two apoptosis-inducing drugs, staurosporine (STS) and 5-fluorouracil (5FU), respectively, to see whether the intracellular $NP_B(Cytc)^*$ would provide any protective effects.

Apoptosis can be visualized directly by fluorescence microscopy when the cells are stained with acridine orange/ethidium bromide (AO/EB)[71]. AO (green) stains both live and dead cells. EB (orange), on the other hand, only stains cells that have lost membrane integrity. Under a fluorescence microscope, live cells are uniformly green; cells at early apoptosis show green dots in the nuclei and those at late apoptosis give orange emission. Figure 7a shows that nonimprinted nanoparticles (NINPs) at 100 μM have no protective effects when apoptosis is induced by STS. In the presence of the same concentration of $NP_B(Cytc)^*$, significantly less apoptosis is observed.

Cytc is found in between the outer and inner membranes of mitochondria of healthy cells and is released from all mitochondria within ~5 min when an apoptotic stimulus is applied[72]. The final concentration of the protein throughout the cell is estimated to be in the range of 5 to 150 μM[73]. Despite the large amounts of Cytc to be targeted, 100 μM of $NP_B(Cytc)^*$ offers significant protection to the cells, indicating that the NP binding is able to compete with the natural binding partners of the protein, inhibiting apoptosis.

The protective effect of $NP_B(Cytc)^*$ was quantified by FACS using apoptosis detection kits (Alexa Fluor® 488 annexin V and propidium iodide). Figure 7b shows that the inhibitory effect of $NP_B(Cytc)^*$ depends on the amounts of STS used to induce cell death. This is a reasonable result because a higher amount of Cytc is expected to be released under a higher drug loading, but the same amount of $NP_B(Cytc)^*$ is used for the apoptosis intervention in all experiments. At a concentration of 0.1 μM of STS, the inhibitory effect of $NP_B(Cytc)^*$ for apoptosis is ca. 33%. Figure 7b also includes a positive control—pepstatin-A (pep A), which is a cathepsin D inhibitor that prevents the release of Cytc[74]. NINP acts as a negative control and offers no protection to the cells. Clearly, the molecular recognition of $NP_B(Cytc)^*$ is essential to its apoptosis-inhibiting effects. Similar observations are made when 5FU is used to induce apoptosis (Fig. 7c).

In summary, proteins are workhorses of cell and their biological functions are regulated frequently by the binding with their partners. We have demonstrated that water-soluble polymeric nanoparticles can be made to recognize a variety of proteins different in PIs and molecular weights. Preparation of the nanoparticle protein-binders is accomplished through a combination of dynamic covalent chemistry and a double imprinting strategy, using trifluoromethyl ketone functional monomers to bind the surface lysines of a protein. The nanoparticles display dissociation constants in the hundreds of nanomolar range and excellent specificity for their targets (Fig. 3a), thanks to the many interactions that define the binding partners, including the distribution of signature lysines on the protein, hydrophobic/electrostatic characteristics around the binding lysines, size/shape of the protein, and distribution of acidic groups near the binding site. Although the method does not allow us to target a predetermined region of a protein surface, it is general and the binding is strong enough to compete with some natural PPIs. With low cytotoxicity and facile entry into cells by both energy-dependent and -independent pathways, these materials could become powerful tools to probe protein functions in vitro and in cellulo.

## Methods

### General procedure for the $NP_A$ preparation

FM **1a** (0.0032 mmol), DMPA (10 μL of a 12.8 mg/mL solution in DMSO, 0.0005 mmol), and MBAm (0.04 mmol) were added to a micellar solution of **3** (9.3 mg, 0.02 mmol) in water (2.0 mL). The reaction mixture was ultrasonicated for 10 min, followed by the addition of **4** (3.4 mg, 0.02 mmol), $CuCl_2$ (10 μL of a 6.7 mg/mL solution in $H_2O$, 0.0005 mmol), and sodium ascorbate (10 μL of a 99 mg/mL solution in $H_2O$, 0.005 mmol). The mixture was stirred slowly at room temperature for 12 h before the protein template (0.0004 mmol) was added. After another 12 h of stirring at room temperature, the mixture was

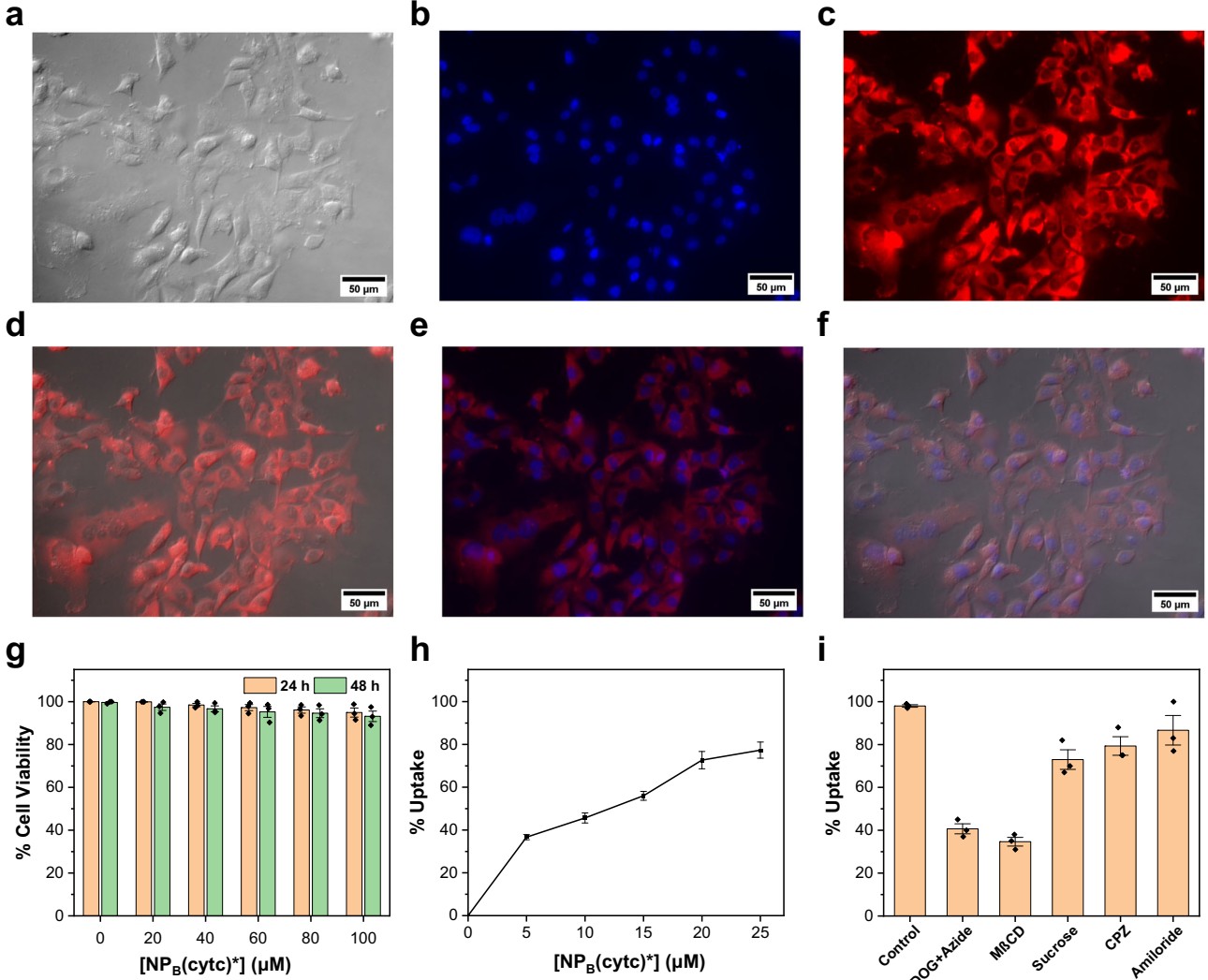

**Fig. 6 | Cell uptake of NP_B(Cytc)*. a–f** Fluorescence imaging of MDA-MB-231 cells incubated 20 μM NP_B(Cytc)*: **a** brightfield image of cells; **b** image of cells in the blue channel (to visualize the nucleus-binding Hoechst 33342); **c** image of cells in the red channel (to visualize NP_B(Cytc)*); **d** overlay of the brightfield image and the red-channels image; **e** overlay of (**b**, **c**); **f** overlay of (**a**–**c**). **g** Cell viability in the MTT assay. **h** Percent uptake of NP_B(Cytc)* by MDA-MB-231 cells as a function of the NP concentration. **i** Percent uptake of NP_B(Cytc)* by MDA-MB-231 cells in the presence of different inhibitors: DOG+Azide 2-deoxy-D-glucose + sodium azide, MβCD methyl-β-cyclodextrin, Sucrose hypertonic sucrose, CPZ chlorpromazine. Data were presented as the mean ± SE, $n = 3$ independent biological samples.

transferred to a glass vial and purged with nitrogen for 15 min. After the sample was irradiated in a Rayonet reactor (300 W/m², 365 nm) for 12 h, **5** (10.6 mg, 0.04 mmol), CuCl₂ (10 μL of a 6.7 mg/mL solution in H₂O, 0.0005 mmol), and sodium ascorbate (10 μL of a 99 mg/mL solution in H₂O, 0.005 mmol) were added. After the mixture was stirred at room temperature for another 6 h, the aqueous reaction mixture was poured into 8 mL of acetone. The precipitate was collected by centrifugation (2500×g for 10 min) and washed with a mixture of acetone/water (3 × 5 mL/1 mL), methanol/acetic acid (3 × 5 mL/0.1 mL), and excess methanol before air-dried. Typical yields for **NP_A** were ~80%. Removal of the template was confirmed by MALDI MS analysis (Supplementary Fig. 8).

### General procedure for the NP_B preparation

FM **1a** (0.0032 mmol), DMPA (10 μL of a 12.8 mg/mL solution in DMSO, 0.0005 mmol), and MBAm (0.04 mmol) were added to a micellar solution of **3** (9.3 mg, 0.02 mmol) in water (2.0 mL). The reaction mixture was ultrasonicated for 10 min, followed by the addition of **4** (3.4 mg, 0.02 mmol), CuCl₂ (10 μL of a 6.7 mg/mL solution in H₂O, 0.0005 mmol), and sodium ascorbate (10 μL of a 99 mg/mL solution in

H₂O, 0.005 mmol). The mixture was stirred slowly at room temperature for 12 h before the protein template (0.0004 mmol) was added. After another 12 h of stirring at room temperature, the mixture was transferred to a glass vial and purged with nitrogen for 15 min. After the sample was irradiated in a Rayonet reactor (300 W/m², 365 nm) for 12 h, compound **6** (0.001 mmol), CuCl₂ (20 μL of a 6.7 mg/mL solution in H₂O, 0.001 mmol), and sodium ascorbate (10 μL of a 99 mg/mL solution in H₂O, 0.01 mmol) were added. The reaction mixture was stirred for an additional 12 h. Subsequently, acrylamide (45 mg, 0.63 mmol), MBAm (15 mg, 0.097 mmol), and compound **7** (1.1 mg, 0.004 mmol) were added, and the reaction mixture was purged with nitrogen for 30 min, sealed with a rubber septum, and irradiated in a Rayonet reactor at room temperature for 30 h under nitrogen. Then the mixture was poured into acetone (30 mL). The precipitate was collected by centrifugation (2500×g for 10 min) and washed with a mixture of acetone/water (5 × 5 mL/0.5 mL), acetone/acetic acid (5 × 5 mL/50 μL), and methanol/water (5 × 5 mL/0.5 mL). The sample was vortexed for 1 min before each centrifugation. The off-white powder was dried under vacuum to afford the final **NP_B** with a typical yield of ~70%.

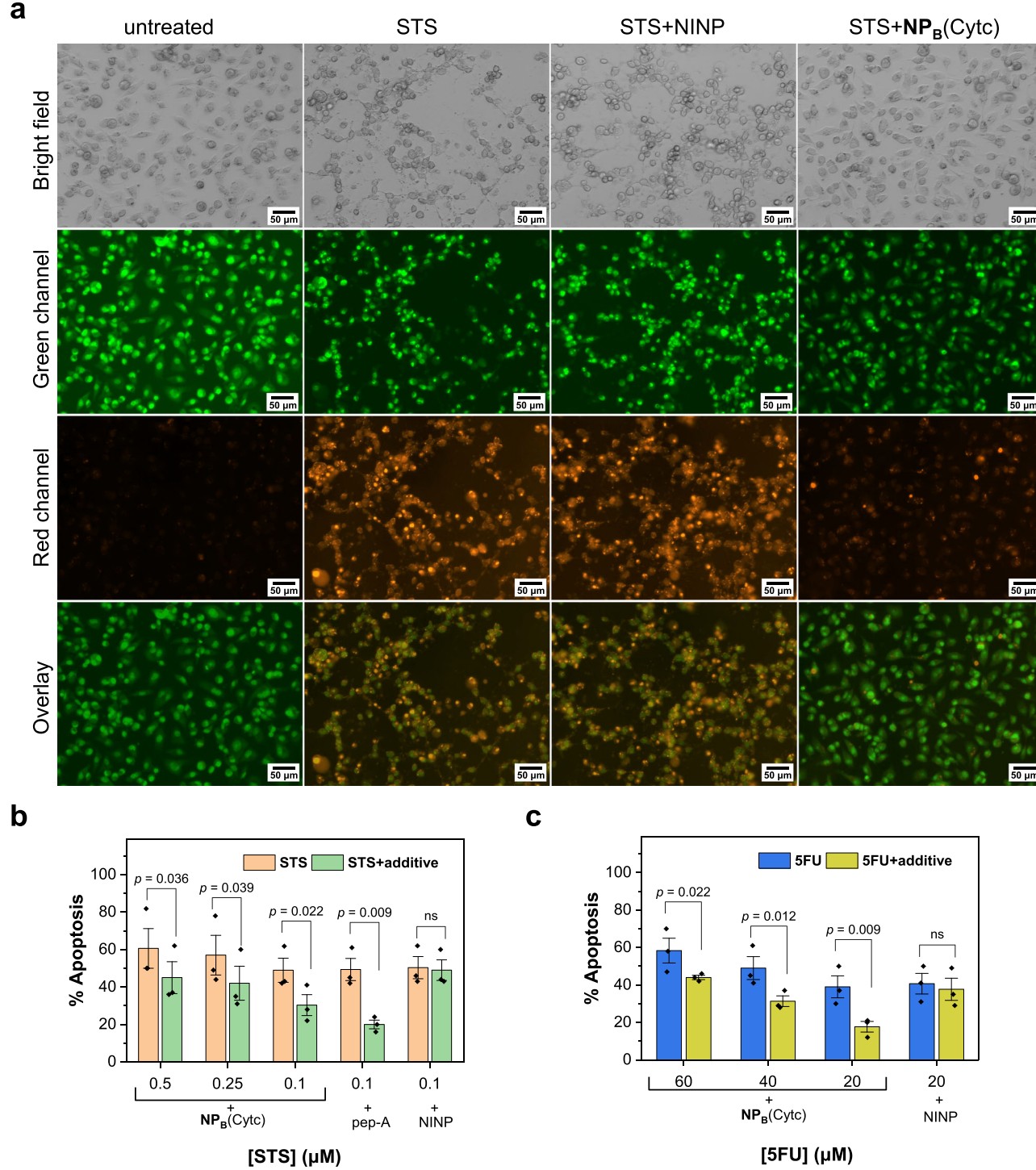

**Fig. 7 | Apoptosis intervention by intracellular NP_B(Cytc)*. a** Monitoring of the apoptosis of MDA-MB-231 cells under different conditions by fluorescence imaging. The cells were stained with acridine orange/ethidium bromide (AO/EB) after treatment with nanoparticles and the apoptosis-inducing STS. **b** Percent apoptosis induced by different concentrations of saurosporine (STS) determined by FACS. Pepstatin-A (pep A) was a positive control protecting cells from apoptosis. **c** Percent apoptosis induced by different concentrations of 5-fluorouracil (5FU) determined by FACS. [NP_B(Cytc)] = [pep A] = [NINP] = 100 μM in the experiments. For the apoptosis data, a linear mixed model was used to compare each treatment with the control at each dosage (Supplementary Tables 11–13 and Supplementary Figs. 26, 27). Data were presented as the mean ± SE, *n* = 3 independent biological samples and *p* values from two-sided *t*-test based on the linear mixed models. ns not significant.

## General procedure for the NP_B(Cytc)* preparation

FM **1a** (0.0032 mmol), DMPA (10 μL of a 12.8 mg/mL solution in DMSO, 0.0005 mmol), and MBAm (0.04 mmol) were added to a micellar solution of **3** (9.3 mg, 0.02 mmol) in water (2.0 mL). The reaction mixture was ultrasonicated for 10 min, followed by the addition of **4** (3.4 mg, 0.02 mmol), CuCl_2 (10 μL of a 6.7 mg/mL solution in H_2O, 0.0005 mmol), and sodium ascorbate (10 μL of a 99 mg/mL solution in H_2O, 0.005 mmol). The mixture was stirred slowly at room temperature for 12 h before Cytc (0.0004 mmol) was added. After another 12 h of stirring at room temperature, the mixture was transferred to a glass

vial and purged with nitrogen for 15 min. After the sample was irradiated in a Rayonet reactor (300 W/m$^2$, 365 nm) for 12 h, compound **8** (0.001 mmol), CuCl$_2$ (20 μL of a 6.7 mg/mL solution in H$_2$O, 0.001 mmol), and sodium ascorbate (10 μL of a 99 mg/mL solution in H$_2$O, 0.01 mmol) was added. The mixture was stirred for 8 h in dark. Afterward, compound **6** (0.001 mmol), CuCl$_2$ (20 μL of a 6.7 mg/mL solution in H$_2$O, 0.001 mmol), and sodium ascorbate (10 μL of a 99 mg/mL solution in H$_2$O, 0.01 mmol) were added. The reaction mixture was allowed to be stirred for an additional 12 h. Next, acrylamide (45 mg, 0.63 mmol), MBAm (15 mg, 0.097 mmol), and compound **7** (1.1 mg, 0.004 mmol) were added. The reaction mixture was purged with nitrogen for 30 min, sealed with a rubber septum, and irradiated in a Rayonet reactor at room temperature for 30 h under nitrogen. The mixture was poured into acetone (30 mL). The precipitate was collected by centrifugation (2500×*g* for 10 min) and washed with a mixture of acetone/water (5 × 5 mL/0.5 mL), acetone/acetic acid (5 × 5 mL/50 μL), and methanol/water (5 × 5 mL/0.5 mL). The sample was vortexed for 1 min before each centrifugation. The light red powder was dried under vacuum to afford the final **NP$_B$**(Cytc)* with a typical yield of ~70%.

## Supplementary Information
Supplementary Methods including general experimental methods, syntheses of small molecules, other experimental details; Supplementary Notes including ITC titration curves, additional data, and statistical analysis of apoptosis data; Supplementary References

## Reporting summary
Further information on research design is available in the Nature Portfolio Reporting Summary linked to this article.

## Data availability
All the data supporting the findings of this study are available within the article and Supplementary Information, or available from the corresponding author on request. Protein structures are from Protein Data Bank (PDB ID: 5IY5 and 1w4w).

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

## Acknowledgements

This work was supported by the National Science Foundation under the awards DMR- 2002659 and 2308625 (Y.Z.). We thank Ms. Lu Liu at Corteva Agriscience for the statistical analysis of the apoptosis data.

## Author contributions

Y.Z. conceived the idea and directed the research; A.G. and M.S. performed the experiments; A.G., M.S., and Y.Z. analyzed the data; Y.Z. wrote the paper.

## Competing interests

The authors declare no competing interests.
