## [Peer Review File · Nature Communications]

Reviewers' Comments:

Reviewer #1:

Remarks to the Author:

The manuscript entitled: "Highly Specific, Cell-Penetrating, Protein-Binding Polymeric Nanoparticles through Dynamic Covalent Chemistry and Double Imprinting" reports a novel method for constructing protein-binding polymeric nanoparticles that can compete with some natural protein-protein interactions, holding potential to become a powerful tool to probe protein functions. This is a very interesting paper, and the reviewer pretty much enjoy the beauty of the imprinting routine design. However, there are some concerns that need to be well addressed before acceptance.

Major comments:

1. Since the whole protein is imprinted, as the authors state that the protein itself is fragile, it is very necessary to verify that the template protein does not undergo drastic conformational changes before and after imprinting, otherwise it will seriously affect the effect of imprinting. Although an aqueous phase system was used in the present study, there were a large number of surfactants and functional monomers after all. In fact, many scientists have turned to do protein imprinting with a characteristic fragment of the target protein as the template (epitope imprinting). The epitope imprinting can well avoid the issues associated with whole protein imprinting and meanwhile provide additional merits such as good availability of templates. Particularly, in recently years, epitope imprinting and its biomedical applications have gained significant progress. Literature citation and discussion on competing approaches are inadequate.
2. The author indirectly claimed that the imprinting factor is >470 . If it is true, this is definitely a super exceptional result. Usually, an imprinting factor more than 10 is already something for protein/peptide imprinting, not to mention a value more than 100! In fact, according to the note in Table 1 (The binding constant could not be determined accurately by ITC due to the weak binding), it is very apparent that such a claim is rather unreasonable due to the uncertainty for the non-imprinted polymer. To confirm such outstanding imprinting effect, more experiments, particularly cross-reactivity to a range of interfering species (structurally close proteins) and binding isotherms for MIP and NIP are necessary.
3. As the same reason for comment #2, Figure 2a fails to support the selectivity!
4. How to control the thickness of the double imprinting layer, different proteins have different sizes, and how to ensure that a more suitable imprinting cavity can be obtained according to different proteins each time. If the layer is too thick, the protein may be embedded; if the layer is too thin, the cross-reactivity will increase. As shown in Figure S5, the shape of the particles is not uniform. Could the thickness be used as a key parameter to get better imprinting performance? If so, how to improve and what effect can be achieved?
5. One may wonder if the presented method is generally applicable. It is very necessary to expand the types of proteins and cell lines for wide verification, beyond only focusing on protein Cytc and MDA-MB-231 cells, thus making this method more general and impactful.

Minor comments:

The description is too technical and needs to be polished so that general readers in more fields such as biology can understand easily.

Reviewer #2:

Remarks to the Author:

This manuscript describes a novel nanoparticle construct that can be templated to bind a specific protein over a large surface area and thus potentially inhibit protein-protein interactions. Starting from a common micellar scaffold, the "double imprinting" strategy involves two stages, each are templated by the target protein: (1) photoinitiated cross-linking in the presence of a functional monomer that is able to form (reversible) covalent interactions with lysine side chains on the protein; (2) photoinitiated cross-linking in the presence of a cationic functional monomer that is proposed to increase the size of the cavity/surface for protein binding and to introduce favourable electrostatic interactions with protein surface carboxylates.

The strategy is, to my knowledge, original. The binding of the imprinted particles is evaluated after

each stage by ITC and shows impressive affinity and selectivity for each targeted protein. The binding affinity increases after each imprinting stage, demonstrating an additive effect of the two-stage procedure.

Potential generality is explored by applying the method to a number of target proteins. Using one of the nanoparticle products, two functional tests are performed: (1) in vitro inhibition of an enzyme-catalysed reaction that relies on electron transfer across a protein complex; (2) in-cell assessment of the ability of particles to inhibit Cytochrome C induced apoptosis.

Altogether this makes for an interesting and novel concept, which targets a general method to tackle a highly challenging and broadly relevant problem. It should be of interest across multiple fields, spanning supramolecular chemistry, nanochemistry the chemical biology. However there are several outstanding questions on the data and issues with the manuscript that would have to be addressed in order to be fully convincing and appropriate for publication.

Major Issues

1. The description of the strategy needs improving. The role of dynamic covalent reactions and the role of each component in the system is not clear from the introduction.

1a. Dynamic covalent chemistry. There is prior art in which dynamic covalent chemistry is used to introduce functional binding units in optimal number and location on a nanoscale scaffold to achieve noncovalent recognition of biomolecules (eg single-chain nanoparticles for lectin binding [10.1002/anie.201706379]; gold nanoparticles for oligonucleotide binding,[10.1002/anie.201409667; 10.1021/acs.langmuir.5b03673] cross-linked polymer nanoparticles for protein recognition[10.1021/ma402402p]). By contrast, the current study uses dynamic covalent bonds as one of the binding interactions between the nanoparticle and the protein (irreversible photo-crosslinking reactions provide the means for anchoring these recognition units in the nanoparticle construct). The second imprinting step optimises noncovalent interactions to improve binding affinity.

1b. Surface-cross-linked micelle. The early introduction discusses "nanoparticles" in general. In paragraph 3, it is written "allows the FM-containing surface-cross-linked micelle (SCM)" with no explanation that the SCM is the "nanoparticle" in question and no explanation as to why a micelle scaffold is used or required. What are the multiple roles of the surfactant molecules and micellar structures?

These distinctions are quite unclear from Scheme 1 and the accompanying introductory description, including the context to the prior art. They only become apparent on reading the full manuscript.

I would encourage the authors to introduce earlier in the manuscript the concept of a two stage process that optimises first (dynamic) covalent interactions and then noncovalent interactions, and to make reference to a wider range of prior art in order to draw attention to the novel aspects of their own study. The double-imprinting concept is a core element of the manuscript, as evident in the title and abstract, so it would be expected that this concept is summarised in a single figure. The role of the nanoparticle scaffold and the reasons for the design chosen when should also be explained when describing the initial strategy.

2. Related to point 1, the description of the details of the method should also be improved. It is confusing that panel (b) of scheme 1 describes preparation of the scaffold required for the imprinting process shown in panel (a). Scheme 1 is dominated by the green "blobs" used to indicate protein but which convey no detailed information. Meanwhile, the molecular structures are poorly presented at low resolution, with inconsistent bond angles and sizes, generally making it difficult to appreciate the molecular basis of the mechanism described. Where molecular structures are also represented by cartoon elements (eg molecule 3) this needs to be better annotated. Many of the same comments also apply to Scheme 2.

3. In describing the construction of the micelle scaffold it is described that cross-linking leaves

behind unfunctionalized alkynes “for further click functionalization”. It is not described whether this was intended, and if so, why this was deemed necessary. Surface functionalization with 5 is not described until p4 and even then it is not clear to me the role that this unit was intended to play. Was this a modification required simply for efficient isolation and processing the particles? The templating protein is still present, so is the introduction of 5 also involved in optimising protein recognition?

4. It should be described (in main text or SI) how NPA/B are isolated from the templating proteins, and how purification is verified. This is critical to ensuring that the measured association constants reflect the imprinted NPs only.

5. Surfactant design. Surfactant 3 is positively charged and so would intuitively be expected to be appropriate for constructing systems that recognize negatively charged proteins. However, the choice and design of surfactant 3 is not explained. This raises a critical question: is this “general” method applicable to recognising both positively and negatively charged proteins?

6. Related to point 5. It would be expected that electrostatics are a crucial factor determining binding affinity and selectivity; but this is hardly discussed in the results (only a very brief comment at the bottom of p7). There are zeta potential measurements for only one pair of imprinted NPs (NPA/B lysozyme) given in the SI but I cannot find mention of these in the main text. Zeta potential measurements should be given for all imprinted NPs, reported in tables 1 and 2 alongside information on the charge state (e.g. PI) of the proteins. This will aid interpretation as to whether the differences in binding affinity are driven by electrostatics. The authors should discuss to what extent the trends in binding affinity can be explained by electrostatics.

7. Contact surface area is a crucial parameter for protein-protein interactions and one of the rationales given for the second imprinting step is an increase in size of the recognition interface. Yet, there is no discussion of NP (or protein) size given. The SI only has DLS results for one pair of NPs. Size information from DLS should be given for all imprinted NPs in the main text, ideally alongside DLS (or at least molecular weight) information for each of the templating proteins. As in the case of point 6, this is essential for (a) understanding the basis of the observed binding affinities and selectivities; and (b) assessing the claimed generality of the method.

8. The detailed information on binding thermodynamics given in tables 1 and 2 is not discussed in the text. The authors should provide some discussion of these results, but I suggest the thermodynamic parameters ΔG , ΔH and ΔS are removed from the main text tables to make way for the information on size and zeta potential.

9. The claims made about the protective effect from CycC-induced apoptosis require a statistical test of significance in order to be valid.

10. The characterization of new organic compounds is incomplete. In particular ^{13}C NMR spectroscopy data is not consistent with the presence of CF_3 groups which should result in a quartet in all ^{13}C NMR spectra. This is not apparent in either the reported data nor the reproduced spectra.

Other points

11. “cross-linking density” is mentioned (p2) as a key parameter for macromolecular templates. How was this optimised in the current study?

12. p3: “the highly efficient click reaction” is imprecise. There are many highly efficient click reactions. The reaction in question is a copper-catalyzed azide-acetylene cycloaddition.

13. The molar ratios of each component were “optimized experimentally” (p4). It is not described how this optimisation was achieved. Based on what target criterion? Were NPs varying each of the components prepared, isolated and tested for binding, or was only the ratio of functional monomer varied? The statement (p5) “Our screening shows that a 8:1 ratio of FM/template gave NPA(lysozyme) the strongest binding for its templating protein” does not make it clear that this

ratio must be optimised for each different functional monomer, as is apparent from Figure 1.

14. There is information in the SI (Figure S1) regarding monitoring of the nanoparticle cross-linking reactions by ¹H NMR spectroscopy. This information and the conclusions drawn needs to be referred to in the main text.

15. A key attribute of dynamic covalent chemistry is that it allows preparation of dynamic combinatorial libraries allowing optimum binding structures can be identified from mixtures. Why did the authors not investigate particles made by mixing more than one functional monomer?

16. Figure 1 caption (and elsewhere). The phrase "Binding constants of NPA for ..." gives the impression that NPA is a single entity. In fact, this figure describes binding constants for several different NPs. This needs to be clear and consistent in the nomenclature used here and throughout the manuscript.

17. p6 A cross reference to Scheme 2 is required when describing the second imprinting step.

18. Table 2. Is it correct that ratio [7]:template is > 1 in most experiments? Entry 1 suggests that this particle was constructed using 7 but no template. The phrase "this outer layer 7 enabled the binding of NPB(lysozyme) to reach $K_a = (11.8 \pm 2.0) \times 10^5 \text{ M}^{-1}$ and increase further with FM 7 to $32.8 \times 10^5 \text{ M}^{-1}$ " does not make it clear where 7 was present, and where it was not present.

19. Information and discussion of relative NP and protein sizes (see point 7) would be useful in informing the currently vague discussion of the different effects of NPA and NPB on HRP function(p9).

20. In discussing cellular entry of NPs (p 11), the authors refer to literature results on sub-10 nm gold nanoparticles (ref 62). Although cationic, these structures likely behave very differently to the large organic particles described in the current manuscript. The authors should defend this comparison or else make reference to systems that are closer in nature to their particles.

21. Conclusions, p11. The phrase "The nanoparticles display hundreds of nanomolecular binding affinities" needs clarified

22. Conclusions. There is nothing in the imprinting strategy that specifically targets the PPI site over any other area on the protein surface. This is a notable feature of the current strategy, which at the same time suggests a limitation but also makes the binding affinities and functional results observed quite remarkable. The authors should provide some discussion of this feature.

23. Reference 33 requires full bibliographic information

24. footnote 38. Numeral 3 should be bold font

Reviewer #3:

Remarks to the Author:

Overall comments

This paper describes an interesting approach to a particularly difficult problem: how to build efficient and general strategies to engineer protein surface recognition with the view of impacting cellular biological and develop novel medicines. On the smaller scale, small molecule can achieve this after extensive rounds of medicinal chemistry. These small molecules are able to disrupt protein-protein interactions but often require the presence of pockets (existing or induce via binding) to establish significantly stable interactions. This approach can be challenging with proteins that lacking druggable features. The approach proposed by the authors in this manuscript potentially provides a solution to this challenge. It is therefore an exciting conceptual advance.

The particular use of the molecular imprinting described for self-organising nanoparticles is a

clever approach. It is a complex but well-engineered system that the authors seem to control relatively well. The paper overall is well written and clear. The supporting methods for the chemistry and biophysical characterisations are relatively well developed. The biological evaluation of the imprinted nanoparticles is less convincing mostly because the lack of statistical repeats impacts the robustness of the conclusions.

While the approach is innovative and seems to work, the binding affinities observed remain in the high micromolar range which won't transferable into a therapeutic outcome.

Finally, one major question after reading this manuscript is the selection of proteins targets. Some of them, such as BSA don't seem particularly relevant from a therapeutic perspective. There is no rationale provided as to their selection for the imprinting. Proteins involved in intrinsic apoptosis or other key biological pathways involving PPIs would have been a lot more relevant to demonstrate the power of the method (BCL-2, BAX/BAK,...). They would have provided strong proof-of-concept data with lots of small molecules to benchmark against.

Specific comments

P2, top of the page "Preorganizing the charged, hydrogen-bonding, and hydrophobic groups over such a large surface area on a synthetic scaffold is an extremely daunting task". While I agree with this comment, it is important to note that small molecules don't need to interact with the entire interface to exhibit tight binding interaction with these large interfaces. There are now many examples of small molecules inhibitor of PPIs with drug like properties. These concepts (e.g. hot-spots, binding epitopes) have been well described in the literature.

There are a few sections of the paper that are unclear:

- I find the notion that the FM small molecules are in an equilibrium between the inside and outside of the SCMs interesting and understandable, especially that it is core to the strategy described. Do the authors have data to support the fact that at steady state, in absence and in the presence of a given protein target, these FMs are mostly within the SCM? While the hydrophobicity of the SCM provides a pull for the FMs, they could still diffuse out in the media over time.

- On page 3, third paragraph "However, 1a is overall hydrophobic. A protein complexed with several FMs thus possesses multiple hydrophobic "anchors" or "tentacles" and should easily latch onto the SCM. Inserting these aromatic ketones into the micelle will not only bury the hydrophobic FMs inside the micelle but also stabilize the hemiaminal by moving them out of water." I am having trouble understanding how the "tentacles" can re-insert into the SCM when reversibly bound to the protein. They presumably need to "extend" far enough to be able to reach beyond the layer of tripropargylammonium head groups. Can the authors provide some indication of scale here, to support the fact that the protein-hemiaminal adduct can actually be inserted into the SCMs.

- How many proteins can one SCM imprint with? Would there be cases where one SCM imprints with than one protein?

- Can the author show that the NP(B) enter the space between inner and outer mitochondrial membranes where cytc resides when it executes its normal, non-apoptotic function? Presumably, this is where the NP(B) would need to go to be able to inhibit the oxidation of Cyt C.

- In the experiments assessing the inhibition of apoptotic in cells, can the authors eliminate the impact on CcO? Is it possible to provide evidence of the disruption (or lack of formation) of the apoptosome by the NP(B).

- The data obtained in the apoptosis inhibition experiments is encouraging although there is a need for very high concentrations to achieve this result. While exciting, these results could still be driven by indirect mechanism of action. One interesting control would be to run an experiment with an NP(B) imprinted for a sperate protein (not just the non-imprinted NP(B))?

- Are the +/- values indicated in the ITC data tables representing standard deviations? Please clarify.

- The methods are appropriately detailed but there seem to be a lack of repeats for a number of biological experiments presented. For the ITC experiments, it is noted in the caption that the experiments have been run in triplicates. The authors should clarify whether this means 3 independent experiments, or one experiment run in triplicate. If the latter, repeats experiments need to be conducted to be able to show any S.D or SEM values.
- In other data sets, there are no SD or SEM values included (e.g. Fig 1, Fig 2, Fig 3, Fig 4, S14) suggesting the experiments have been run only once. For robustness, these experiments should be also repeated.
- Fig 5g and 5i: what are the error bars representing? Please indicate whether this data arises from independent repeats.
- Figure 6b and c, Fig S8, S9: here again, indicate what the error bars are representing and whether these experiments have been run several times (in independent runs). The error bars are very small for a biological experiment in cells.

Minor comments:

Some references are little old and do not integrate the latest thinking around PPIs (e.g. references 7-9).

P9, second last paragraph "...can strongly impact the electron transfer...": the concentrations used to achieve these effects are quite high so I wouldn't characterise this effect as "strong".

The word "significant" is used several times across the manuscript. Yet, there is no statistical analysis to determine whether the difference between results is actually statistically significant.

The word concentration is misspelled in most of the graphs included in the supplementary info section.

In response to reviewer 1:

The manuscript entitled: “Highly Specific, Cell-Penetrating, Protein-Binding Polymeric Nanoparticles through Dynamic Covalent Chemistry and Double Imprinting” reports a novel method for constructing protein-binding polymeric nanoparticles that can compete with some natural protein–protein interactions, holding potential to become a powerful tool to probe protein functions. This is a very interesting paper, and the reviewer pretty much enjoy the beauty of the imprinting routine design. However, there are some concerns that need to be well addressed before acceptance.

- We thank the reviewer for the positive feedback on our work.

Major comments:

1. Since the whole protein is imprinted, as the authors state that the protein itself is fragile, it is very necessary to verify that the template protein does not undergo drastic conformational changes before and after imprinting, otherwise it will seriously affect the effect of imprinting. Although an aqueous phase system was used in the present study, there were a large number of surfactants and functional monomers after all. In fact, many scientists have turned to do protein imprinting with a characteristic fragment of the target protein as the template (epitope imprinting). The epitope imprinting can well avoid the issues associated with whole protein imprinting and meanwhile provide additional merits such as good availability of templates. Particularly, in recently years, epitope imprinting and its biomedical applications have gained significant progress. Literature citation and discussion on competing approaches are inadequate.

- Surfactants are indeed often incompatible with proteins. As mentioned in footnote 44, in our hands, mixing **3** and proteins in aqueous solution frequently gives a precipitate.
- As suggested, we recorded the CD spectrum of our protein template (lysozyme) by itself and in the presence of the functional monomer (**1a**) and SCM. Figure S2 shows negligible changes in the CD signals, suggesting that the SCM and FM are compatible with the protein and not changing its conformation. As discussed in the main text, this is mainly because SCMs with their hydrophobic tails tucked inside have very little surface activity.
- Figure S2 also shows that the CD signals for the protein after photopolymerization of the micelle core, again showing little changes in the peak shapes. This suggests that the protein retains its conformation during the entire imprinting process.
- We recognize that epitope imprinting is an alternative approach to whole-protein imprinting and have discussions on the merits and limitations of the two approaches. In response to the comment, we added several recent epitope imprinting papers to the reference and highlighted the benefits of epitope imprinting in the revised manuscript.

2. The author indirectly claimed that the imprinting factor is >470 . If it is true, this is definitely a super exceptional result. Usually, an imprinting factor more than 10 is already something for protein/peptide imprinting, not to mention a value more than 100! In fact, according to the note in Table 1 (The binding constant could not be determined accurately by ITC due to the weak binding), it is very apparent that such a claim is rather unreasonable due to the uncertainty for the non-imprinted polymer. To confirm such outstanding imprinting effect, more experiments, particularly cross-reactivity to a range of interfering species (structurally close proteins) and binding isotherms for MIP and NIP are necessary.

3. As the same reason for comment #2, Figure 2a fails to support the selectivity!

- A $1200\text{ M}^{-1} K_a$ value was used as the higher limit for the binding between NINP and lysozyme, based on our ITC titration. This value was then used to calculate the imprinting factor for NP_A (lysozyme), ~ 470 . The reviewer’s point is well-taken in that there is uncertainty in the

measurement for the binding constant for the NINP. Proteins aggregate easily at higher concentrations, making it difficult to measure low-affinity binding constants. However, a binding constant of over 1000 M^{-1} is typically measurable by ITC but our titration showed no binding at all (Figure S9f). Thus, we consider using a K_a of 1200 M^{-1} is reasonable for the estimation.

- Another way of estimating the imprinting effect is to compare entries 1 and 7 in Table 1. The nanoparticle prepared without any FM has a weaker binding that is high enough to be measurable (Figure S9g). If this number of 9200 M^{-1} is used for the NINP binding, the imprinting factor is still ~ 60 . The real imprinting factor must be much higher because the value of 9200 M^{-1} (Table 1, entry 7) is for an imperfectly imprinted nanoparticle, not nonimprinted nanoparticle.
- Because of the difficulty in measuring low-affinity binding constants for proteins (due to their aggregation at high concentrations), there is also uncertainty in the binding selectivities depicted in Figures 1a and 2a, as pointed out by the reviewer. In the revision, we measured the binding between lysosome and NP_A (lysozyme) in the presence of 2 equivalents of a protein competitor (BSA, HRP, α -amylase, cytochrome C, chymotrypsin, OVA, transferrin, and trypsin). The binding constants obtained have an average of $K_a = (53.7 \pm 8.4) \times 10^4 \text{ M}^{-1}$ (Table S2, Figure S11), which is experimentally the same as the binding constant of $K_a = (57.3 \pm 2.8) \times 10^4 \text{ M}^{-1}$ for the binding without the protein competitors (Table 1, entry 1). Thus, the presence of other proteins at twice the concentration has no noticeable effects on the desired binding between lysozyme and NP_A (lysozyme). This also strongly supports the selectivity of the nanoparticles.
- Another piece of evidence for the selectivity of the binding, albeit indirect, comes from the intracellular activity of NP_B (Cytc) (Figure 6). A cell contains thousands and more competing proteins at an overall concentration much higher than what can be achieved in the ITC experiments. Yet, the nanoparticles could selectively inhibit the Cytc-induced apoptosis, suggesting that the binding is highly specific.
- In the revision, we briefly discussed these points to help the reader assess the situations.

4. How to control the thickness of the double imprinting layer, different proteins have different sizes, and how to ensure that a more suitable imprinting cavity can be obtained according to different proteins each time. If the layer is too thick, the protein may be embedded; if the layer is too thin, the cross-reactivity will increase. As shown in Figure S5, the shape of the particles is not uniform. Could the thickness be used as a key parameter to get better imprinting performance? If so, how to improve and what effect can be achieved?

- This is an excellent point. The photo RAFT polymerization is slow (ref 47), taking about 30 hours to complete. The thickness of the outer polymer layer can be controlled by the time of irradiation.
- We likewise were concerned that too thick a polymer layer would bury the templating protein in the 2nd imprinting. However, we found the particle size plateaued to 10–11 nm even if large amounts of acrylamide (AM) and MBAM were used. Our hypothesis is that, once a certain thickness is reached, the AM/MBAM molecules in the solution have difficulty diffusing into the cross-linked polymer layer to reach the photo-activated radicals to be further polymerized. The current preparation was experimentally optimized for the highest binding.
- DLS shows a fairly narrow distribution of particle size (Figures S4 and S5). Because these organic nanoparticles have low electron-density, they have to be negatively stained before TEM imaging. In our experience, these cationic nanoparticles aggregate strongly during the staining and the aggregation is mainly responsible for the non-uniformity observed in the TEM images.

5. One may wonder if the presented method is generally applicable. It is very necessary to expand the types of proteins and cell lines for wide verification, beyond only focusing on protein Cytc and MDA-MB-231 cells, thus making this method more general and impactful.

- We strive to demonstrate the generality in the method of making protein-binding nanoparticles in this work. We believe this goal has been achieved through the molecular imprinting of proteins with various M.W. and PI (Table S1).
- The biological applications are meant to be proof-of-concept (partly because our chemistry-focused lab is not equipped to do in-depth biological research). We mostly intend to demonstrate two points in these experiments. First, the binding is strong enough to inhibit certain PPIs in vitro. Second, the nanoparticles are able to go inside cells to bind their protein targets and intervene PPIs.
- We are in the process of collaborating with biologists on using these nanoparticles for biological applications and will publish the work in due course.

Minor comments:

The description is too technical and needs to be polished so that general readers in more fields such as biology can understand easily.

- We appreciate the comment. Reviewer 2 has many specific suggestions for us to increase the clarity of the technical description of the method. We will address them point-by-point in the following sections..

In response to reviewer 2:

This manuscript describes a novel nanoparticle construct that can be templated to bind a specific protein over a large surface area and thus potentially inhibit protein-protein interactions. Starting from a common micellular scaffold, the “double imprinting” strategy involves two stages, each are templated by the target protein: (1) photoinitiated cross-linking in the presence of a functional monomer that is able to form (reversible) covalent interactions with lysine side chains on the protein; (2) photoinitiated cross-linking in the presence of a cationic functional monomer that is proposed to increase the size of the cavity/surface for protein binding and to introduce favourable electrostatic interactions with protein surface carboxylates.

The strategy is, to my knowledge, original. The binding of the imprinted particles is evaluated after each stage by ITC and shows impressive affinity and selectivity for each targeted protein. The binding affinity increases after each imprinting stage, demonstrating an additive effect of the two-stage procedure.

Potential generality is explored by applying the method to a number of target proteins. Using one of the nanoparticle products, two functional tests are performed: (1) in vitro inhibition of an enzyme-catalysed reaction that relies on electron transfer across a protein complex; (2) in-cell assessment of the ability of particles to inhibit Cytochrome C induced apoptosis.

Altogether this makes for an interesting and novel concept, which targets a general method to tackle a highly challenging and broadly relevant problem. It should be of interest across multiple fields, spanning supramolecular chemistry, nanochemistry the chemical biology. However there are several outstanding questions on the data and issues with the manuscript that would have to be addressed in order to be fully convincing and appropriate for publication.

- We thank the reviewer for the positive feedback on the novelty of the concept and significance of the results.

Major Issues

1. The description of the strategy needs improving. The role of dynamic covalent reactions and the role of

each component in the system is not clear from the introduction.

1a. Dynamic covalent chemistry. There is prior art in which dynamic covalent chemistry is used to introduce functional binding units in optimal number and location on a nanoscale scaffold to achieve noncovalent recognition of biomolecules (eg single-chain nanoparticles for lectin binding [10.1002/anie.201706379]; gold nanoparticles for oligonucleotide binding,[10.1002/anie.201409667; 10.1021/acs.langmuir.5b03673] cross-linked polymer nanoparticles for protein recognition[10.1021/ma402402p]). By contrast, the current study uses dynamic covalent bonds as one of the binding interactions between the nanoparticle and the protein (irreversible photo-crosslinking reactions provide the means for anchoring these recognition units in the nanoparticle construct). The second imprinting step optimises noncovalent interactions to improve binding affinity.

1b. Surface-cross-linked micelle. The early introduction discusses “nanoparticles” in general. In paragraph 3, it is written “allows the FM-containing surface-cross-linked micelle (SCM)” with no explanation that the SCM is the “nanoparticle” in question and no explanation as to why a micelle scaffold is used or required. What are the multiple roles of the surfactant molecules and micellar structures?

These distinctions are quite unclear from Scheme 1 and the accompanying introductory description, including the context to the prior art. They only become apparent on reading the full manuscript.

I would encourage the authors to introduce earlier in the manuscript the concept of a two stage process that optimises first (dynamic) covalent interactions and then noncovalent interactions, and to make reference to a wider range of prior art in order to draw attention to the novel aspects of their own study. The double-imprinting concept is a core element of the manuscript, as evident in the title and abstract, so it would be expected that this concept is summarised in a single figure. The role of the nanoparticle scaffold and the reasons for the design chosen when should also be explained when describing the initial strategy.

- We thank the reviewer for the very helpful suggestions. In the revision, we combined the two schemes into one and introduced the double imprinting strategy in the beginning of the results and discussion.
- We clarified the relationship between cross-linked micelles and nanoparticles early in the discussion.
- We also added several references to the application of the dynamic covalent chemistry in biomolecular recognition. We thank the reviewer for bringing these papers to our attention.

2. Related to point 1, the description of the details of the method should also be improved. It is confusing that panel (b) of scheme 1 describes preparation of the scaffold required for the imprinting process shown in panel (a). Scheme 1 is dominated by the green “blobs” used to indicate protein but which convey no detailed information. Meanwhile, the molecular structures are poorly presented at low resolution, with inconsistent bond angles and sizes, generally making it difficult to appreciate the molecular basis of the mechanism described. Where molecular structures are also represented by cartoon elements (eg molecule 3) this needs to be better annotated. Many of the same comments also apply to Scheme 2.

- We modified the Schemes and annotated structures better, as suggested by the reviewer. Scheme 1 now has two parts, illustrating the 1st and the 2nd molecular imprinting. Everything including the cross-linkable surfactant is annotated. For example, the following annotations are added to the caption: “The red and blue spheres on the protein represent positively and negatively charged surface groups. The magenta spheres on NP_B represent carboxylate-binding functional groups installed through polymerization of FM 7.”

- The word file has very clear graphics. The low-resolution images in the pdf file are likely caused by the pdf conversion.

3. In describing the construction of the micelle scaffold it is described that cross-linking leaves behind unfunctionalized alkynes “for further click functionalization”. It is not described whether this was intended, and if so, why this was deemed necessary. Surface functionalization with **5** is not described until p4 and even then it is not clear to me the role that this unit was intended to play. Was this a modification required simply for efficient isolation and processing the particles? The templating protein is still present, so is the introduction of **5** also involved in optimising protein recognition?

- In the revised Scheme 1, the surface ligands are annotated and added to surface of **NP_A**. These polyhydroxylated ligands help the nanoparticles precipitate from acetone at the end of the reaction and remain insoluble during solvent washing. The protein templates are removed during the process. The surface ligands are mainly added to **NP_A** for enhanced surface hydrophilicity and facile purification. We don't have any evidence that they can enhance the protein binding.
- Alternatively, the cross-linked micelles can be functionalized on the surface by other azide-containing ligands such as **6** in Scheme 1b or **8** for fluorescent labeling.
- In the revision, we clarified these points at the end of the “Design and Synthesis of Protein-Binding NPs” section.

4. It should be described (in main text or SI) how **NPA/B** are isolated from the templating proteins, and how purification is verified. This is critical to ensuring that the measured association constants reflect the imprinted NPs only.

- The nanoparticles are prepared in aqueous solution and purified by precipitating the reaction mixture from acetone and washing the nanoparticles with multiple solvent mixtures (acetone/water three times, methanol/acetic acid three times, and then excess methanol). Removal of the protein template was confirmed by MALDI MS analysis (Figure S3).
- The above information was added to the revised SI.

5. Surfactant design. Surfactant **3** is positively charged and so would intuitively be expected to be appropriate for constructing systems that recognize negatively charged proteins. However, the choice and design of surfactant **3** is not explained. This raises a critical question: is this “general” method applicable to recognising both positively and negatively charged proteins?

6. Related to point 5. It would be expected that electrostatics are a crucial factor determining binding affinity and selectivity; but this is hardly discussed in the results (only a very brief comment at the bottom of p7). There are zeta potential measurements for only one pair of imprinted NPs (**NPA/B** lysozyme) given in the SI but I cannot find mention of these in the main text. Zeta potential measurements should be given for all imprinted NPs, reported in tables 1 and 2 alongside information on the charge state (e.g. **PI**) of the proteins. This will aid interpretation as to whether the differences in binding affinity are driven by electrostatics. The authors should discuss to what extent the trends in binding affinity can be explained by electrostatics.

- We added Table S1 in the SI to summarize the molecular weights and **PI**s of the proteins studied in this work. We also included in the main text Table 3 to summarize all the sizes and zeta potentials for our nanoparticle hosts. In response to the above comments, we added the following discussion to the revised manuscript:
- It should be noted that the protein templates studied include both highly acidic ones (amylase with **PI** \approx 3.5 and BSA with **PI** \approx 5) and basic ones (lysozyme and CytC with **PI** \approx 11), with M.W. ranging from 12 to 80 KDa (Table S1). Yet, Figures 1a and 2a demonstrate that all of them could interact selectively with their corresponding nanoparticle hosts. The zeta potentials of our protein-

binding nanoparticles prepared range from 25–41 mV (Table 3). Apparently, generic electrostatic interactions do not play a large role in the binding. These cross-linked micelles are rigid polymeric nanoparticles with extensive cross-linking. The negative charges on a negatively charged protein are generally distributed over the surface of the protein instead of being concentrated in one region. The generic electrostatic interactions between the protein surface charges and our nanoparticles apparently are unable to compete with the lysine–trifluoromethyl ketone covalent interactions, as supported by Figure 1b.

7. Contact surface area is a crucial parameter for protein-protein interactions and one of the rationales given for the second imprinting step is an increase in size of the recognition interface. Yet, there is no discussion of NP (or protein) size given. The SI only has DLS results for one pair of NPs. Size information from DLS should be given for all imprinted NPs in the main text, ideally alongside DLS (or at least molecular weight) information for each of the templating proteins. As in the case of point 6, this is essential for (a) understanding the basis of the observed binding affinities and selectivities; and (b) assessing the claimed generality of the method.

- We could not add the size information to Tables 1 and 2, because they describe the binding affinity and selectivity of lysozyme-imprinted nanoparticles only.
- As mentioned above, the protein M.W. and PI data are summarized in Table S1 and the sizes and zeta potentials of our nanoparticles in Table 3. Figures 1b and 2b show the correlation of the binding affinities with the surface lysines for NP_A and acidic residues for NP_B . We did not find any correlation between the protein size and binding affinity, as shown by Figures 1a and 2a.

8. The detailed information on binding thermodynamics given in tables 1 and 2 is not discussed in the text. The authors should provide some discussion of these results, but I suggest the thermodynamic parameters ΔG , ΔH and ΔS are removed from the main text tables to make way for the information on size and zeta potential.

- We added the following discussion on the binding thermodynamics:
- The bindings between lysozyme and all the FM-containing NPs are enthalpically derived, with negative (unfavorable) binding entropies (Table 1, entries 1–5). Interestingly, without any FM, the binding is driven by both enthalpic and entropic forces (entry 6). This is a reasonable result given that reversible covalent bonds are mainly responsible for the binding in the trifluoromethyl ketone-containing NPs.

9. The claims made about the protective effect from Cytc-induced apoptosis require a statistical test of significance in order to be valid.

- The statistical analysis is found the revised SI. The following statements are added to the Figure 6 caption:
- The apoptosis experiments were repeated with three independent biological samples. The error bars in Figure 6b,c represent measurement errors in a single experiment. Statistical analysis for the biological replicates is performed using a linear mixed model and reported in the Supporting Information (Tables S11–13 and Figures S19–20). The analysis indicates that $\text{NP}_B(\text{Cytc})$ and pep A significantly lowers the apoptosis level in comparison to the untreated samples at each dosage (p -value < 0.05). NINP, on the other hand, exhibits no significant difference from the untreated sample (p -value = 0.88 and 0.52 for the STS- and 5FU-treated cells, respectively).

10. The characterization of new organic compounds is incomplete. In particular ^{13}C NMR spectroscopy data is not consistent with the presence of CF_3 groups which should result in a quartet in all ^{13}C NMR

spectra. This is not apparent in either the reported data nor the reproduced spectra.

- We apologize for not listing the CF₃ splittings in the previous SI and have corrected the mistakes.

Other points

11. “cross-linking density” is mentioned (p2) as a key parameter for macromolecular templates. How was this optimised in the current study?

- NP_A is cross-linked on the surface through diazide **4** via click chemistry and also by MBAm via free radical polymerization. We clarified the optimization in the revised manuscript: Each surfactant has 3 propargyl groups and each azide cross-linker 2 azides; a 1:1 ratio of [3]/[4] means one equivalent of surface alkynes will be left for further click functionalization if the click cross-linking happens perfectly. The amounts of MBAm and **1a** were optimized experimentally for highest binding constants achievable.

12. p3: “the highly efficient click reaction” is imprecise. There are many highly efficient click reactions. The reaction in question is a copper-catalyzed azide–acetylene cycloaddition.

- We made the suggested changes.

13. The molar ratios of each component were “optimized experimentally” (p4). It is not described how this optimisation what achieved. Based on what target criterion? Were NPs varying each of the components prepared, isolated and tested for binding, or was only the ratio of functional monomer varied? The statement (p5) “Our screening shows that a 8:1 ratio of FM/template gave NPA(lysozyme) the strongest binding for its templating protein” does not make it clear that this ratio must be optimised for each different functional monomer, as is apparent from Figure 1.

- We clarified the optimization in the revised manuscript:
- The NP was prepared with a formulation of [3]/[4]/[5]/[MBAm]/[1a]/[lysozyme] = 50:50:100:100:8:1. We typically choose a surfactant/template of ratio of 50:1 in our preparation because each cross-linked SCM contains approximately 50 surfactants according to dynamic light scattering (DLS).⁵⁴ Each surfactant has 3 propargyl groups and each azide cross-linker 2 azides; a 1:1 ratio of [3]/[4] means one equivalent of surface alkynes will be left for further click functionalization if the click cross-linking happens perfectly. The amounts of MBAm and **1a** were optimized experimentally for highest binding constants achievable. The surface ligand (**5**) is generally used in excess so that the final NP with as many of this ligand on the surface can be precipitated from acetone. Lysozyme is known to have 6 reactive lysines.⁵⁵ Our screening shows that a 8:1 ratio of FM/template gave NP_A(lysozyme) the strongest binding for its templating protein.

14. There is information in the SI (Figure S1) regarding monitoring of the nanoparticle cross-linking reactions by ¹H NMR spectroscopy. This information and the conclusions drawn needs to be referred to in the main text.

- We discussed this point briefly in the main text.

15. A key attribute of dynamic covalent chemistry is that it allows preparation of dynamic combinatorial libraries allowing optimum binding structures can be identified from mixtures. Why did the authors not

investigate particles made by mixing more than one functional monomer?

- We did try mixtures of functional monomers during the early stage of the investigation and found **1a** consistently outperformed the mixtures.

16. Figure 1 caption (and elsewhere). The phrase “Binding constants of NPA for ...” gives the impression that NPA is a single entity. In fact, this figure describes binding constants for several different NPs. This needs to be clear and consistent in the nomenclature used here and throughout the manuscript.

- We made the suggested changes.

17. p6 A cross reference to Scheme 2 is required when describing the second imprinting step.

- As mentioned above, we now combined Schemes 1 and 2 into one Scheme in the revision.

18. Table 2. Is it correct that ratio [7]:template is > 1 in most experiments? Entry 1 suggests that this particle was constructed using 7 but no template. The phrase “this outer layer 7 enabled the binding of NPB(lysozyme) to reach $K_a = (11.8 \pm 2.0) \times 10^5 \text{ M}^{-1}$ and increase further with FM 7 to $32.8 \times 10^5 \text{ M}^{-1}$ ” does not make it clear where 7 was present, and where it was not present.

- We thank the reviewer for catching the typo. The ratio should be for “template/[7]”. Template is always present except for the nonimprinted nanoparticle (NINP). It was the amount of FM 7 that was optimized in these experiments.

19. Information and discussion of relative NP and protein sizes (see point 7) would be useful in informing the currently vague discussion of the different effects of NPA and NPB on HRP function(p9).

- We made the suggested changes and pointed out the thickness of the polymer layer in the revision: **NP_B**(HRP), on the other hand, has an additional polymer layer 2–3 nm thick according to Table 3, surrounding the bound HRP.

20. In discussing cellular entry of NPs (p 11), the authors refer to literature results on sub-10 nm gold nanoparticles (ref 62). Although cationic, these structures likely behave very differently to the large organic particles described in the current manuscript. The authors should defend this comparison or else make reference to systems that are closer in nature to their particles.

- We revised the sentence and the reference to make it more accurate: NPs 10–100 nm in size usually are taken up by cells via energy-dependent endocytosis.⁶⁷ Small cationic nanoparticles (5–10 nm⁶⁸ or even larger⁶⁹), however, can enter cells through direct membrane penetration.

21. Conclusions, p11. The phrase “The nanoparticles display hundreds of nanomolecular binding affinities” needs clarified

- These are calculated from the measured binding constants of **NP_B** shown in Figure 2a. It is clarified in the revision.

22. Conclusions. There is nothing in the imprinting strategy that specifically targets the PPI site over any other area on the protein surface. This is a notable feature of the current strategy, which at the same time suggests a limitation but also makes the binding affinities and functional results observed quite remarkable.

The authors should provide some discussion of this feature.

- We made the suggested changes in the conclusion and noted that the method does not allow us to target specific regions of a protein surface.

23. Reference 33 requires full bibliographic information

- We made the suggested changes.

24. footnote 38. Numeral 3 should be bold font

- We made the suggested changes.

In response to reviewer 3:

This paper describes an interesting approach to a particularly difficult problem: how to build efficient and general strategies to engineer protein surface recognition with the view of impacting cellular biological and develop novel medicines. On the smaller scale, small molecule can achieve this after extensive rounds of medicinal chemistry. These small molecules are able to disrupt protein-protein interactions but often require the presence of pockets (existing or induce via binding) to establish significantly stable interactions. This approach can be challenging with proteins that lacking druggable features. The approach proposed by the authors in this manuscript potentially provides a solution to this challenge. It is therefore an exciting conceptual advance.

The particular use of the molecular imprinting described for self-organising nanoparticles is a clever approach. It is a complex but well-engineered system that the authors seem to control relatively well. The paper overall is well written and clear. The supporting methods for the chemistry and biophysical characterisations are relatively well developed. The biological evaluation of the imprinted nanoparticles is less convincing mostly because the lack of statistical repeats impacts the robustness of the conclusions.

While the approach is innovative and seems to work, the binding affinities observed remain in the high micromolar range which won't transferable into a therapeutic outcome.

Finally, one major question after reading this manuscript is the selection of proteins targets. Some of them, such as BSA don't seem particularly relevant from a therapeutic perspective. There is no rationale provided as to their selection for the imprinting. Proteins involved in intrinsic apoptosis or other key biological pathways involving PPIs would have been a lot more relevant to demonstrate the power of the method (BCL-2, BAX/BAK,...). They would have provided strong proof-of-concept data with lots of small molecules to benchmark against.

- We thank the reviewer for the positive feedback on the novelty of the concept and significance of the results. We would like to point out the binding affinities of NP_A are generally 2–3 μM (Table 1, Figure 1a) and those for NP_B ~300 nM (Table 2, Figure 2a).
- The current work focuses on the developments of materials for protein recognition. The immediate application may not be in therapeutics but for elucidation of biological mechanisms, using the nanoparticles to inhibit/intervene certain PPIs.
- For the above reasons, the protein targets chosen are for the demonstration of protein binding and thus are mostly readily available proteins. Our lab is not set up for in-depth biological work. For those, we would need to collaborate with cell biologists.

Specific comments

P2, top of the page “Preorganizing the charged, hydrogen-bonding, and hydrophobic groups over such a large surface area on a synthetic scaffold is an extremely daunting task”. While I agree with this comment, it is important to note that small molecules don’t need to interact with the entire interface to exhibit tight binding interaction with these large interfaces. There are now many examples of small molecules inhibitor of PPIs with drug like properties. These concepts (e.g. hot-spots, binding epitopes) have been well described in the literature.

- In response, we modified the sentence a little bit, so that it becomes a general statement instead of implying that we have to interact with the entire binding interface: Generally speaking, preorganizing a large number of charged, hydrogen-bonding, and hydrophobic groups over an extended surface area on a synthetic scaffold is an extremely daunting task.

There are a few sections of the paper that are unclear:

- I find the notion that the FM small molecules are in an equilibrium between the inside and outside of the SCMs interesting and understandable, especially that it is core to the strategy described. Do the authors have data to support the fact that at steady state, in absence and in the presence of a given protein target, these FMs are mostly within the SCM? While the hydrophobicity of the SCM provides a pull for the FMs, they could still diffuse out in the media over time.

- These FMs are hydrophobic and have very low solubility in water. They only become soluble with the help of the micelles. Thus, thermodynamically they prefer the interior of micelles. However, unless they have zero solubility in water, there will always be some molecules coming out of the micelles and in a dynamic equilibrium with those inside. This is a generally accepted picture for micelle-solubilized hydrophobic molecules (Myers, D., *Surfactant science and technology*. 2nd ed.; VCH: New York, 1992; Chapter 4.)

- On page 3, third paragraph “However, **1a** is overall hydrophobic. A protein complexed with several FMs thus possesses multiple hydrophobic “anchors” or “tentacles” and should easily latch onto the SCM. Inserting these aromatic ketones into the micelle will not only bury the hydrophobic FMs inside the micelle but also stabilize the hemiaminal by moving them out of water.” I am having trouble understanding how the “tentacles” can re-insert into the SCM when reversibly bound to the protein. They presumably need to “extend” far enough to be able to reach beyond the layer of tripropargylammonium head groups. Can the authors provide some indication of scale here, to support the fact that the protein-hemiaminal adduct can actually be inserted into the SCMs.

- This is a really good question. We modified the description to better present the picture:
- Our hypothesis is that, in the presence of a protein template, **1a** will diffuse out of the SCM to interact with the reactive surface lysines. Meanwhile, hydrophobic voids will be left behind inside the SCM. For a dynamic, noncovalently stabilized micelle, the surfactants will rearrange to eliminate such hydrophobic voids that would have to be filled with water molecules. For a cross-linked micelle, the rearrangement is more difficult, creating a hydrophobic driving force for the FMs to reenter. Hemiaminal easily hydrolyzes in aqueous solution^{46, 47} and the **1a**-protein complex is not expected to be stable in water. However, with the voids “pulling” the protein-conjugated FMs back into the SCM, the protein is pulled together to the SCM, driven by hydrophobic interactions. Once inserted back into the micelle, the hemiaminals are shielded from hydrolysis. From the perspective of the protein template, after reacting with the FMs, it becomes equipped with several hydrophobic “anchors” or “tentacles” to bind to the SCM, ready for imprinting.

- How many proteins can one SCM imprint with? Would there be cases where one SCM imprints with (more) than one protein?

- We typically choose a surfactant/template of ratio of 50:1 in our preparation because each cross-linked SCM contains approximately 50 surfactants according to dynamic light scattering (DLS). In this way, each nanoparticle on average contains a single binding site. This feature is supported by the ITC studies that generally show the number of binding site ~ 1 in the titrations.

- Can the author show that the NP(B) enter the space between inner and outer mitochondrial membranes where cytc resides when it executes its normal, non-apoptotic function? Presumably, this is where the NP(B) would need to go to be able to inhibit the oxidation of Cyt C.

- We are sorry about the confusion. We used a commercially available Cytochrome c Oxidase Assay Kit and employed soluble cytochrome c oxidase to test the activity. Decrease in absorbance of at 550 nm comes from the oxidation of ferrocytochrome c to ferricytochrome c. (https://www.sigmaaldrich.com/US/en/product/sigma/cytocox1?gclid=CjwKCAiA1MCRbAoEiwAC2d64avThFKf-ljZUbd1NlxR6RUDfL5L0Jeu9Zpn2qoJB2X-i25s7NBRjRoCYRwQAvD_BwE).
- We clarified this point in the revision.

- In the experiments assessing the inhibition of apoptotic in cells, can the authors eliminate the impact on CcO? Is it possible to provide evidence of the disruption (or lack of formation) of the apoptosome by the NP(B).

- Cytochrome C-triggered apoptosis is a well-established biological pathway. Ref 57 and 58—e.g., Cytochrome C-mediated apoptosis. *Annu. Rev. Biochem.* **73**, 87-106 (2004); Cytochrome c: functions beyond respiration. *Nat. Rev. Mol. Cell Biol.* **9**, 532-542 (2008)—are two representative reviews on this topic. During this process, Cytochrome C is released from mitochondria to cytosol and no longer associated with CcO. The final concentration of the protein throughout the cell is estimated to be in the range of 5 to 150 μM (ref 76).
- Apoptosome is a very large protein complex formed under specific conditions. Very few labs have the capability to study them in isolated forms. Cytochrome C-triggered apoptosis is well accepted to be mediated by apoptosome, as illustrated in the above mentioned reviews

- The data obtained in the apoptosis inhibition experiments is encouraging although there is a need for very high concentrations to achieve this result. While exciting, these results could still be driven by indirect mechanism of action. One interesting control would be to run an experiment with an NP(B) imprinted for a sperate protein (not just the non-imprinted NP(B))?

- The statistical analysis is found the revised SI. The following statements are added to the Figure 6 caption:
- The apoptosis experiments were repeated with three independent biological samples. The error bars in Figure 6b,c represent measurement errors in a single experiment. Statistical analysis for the biological replicates is performed using a linear mixed model and reported in the Supporting Information (Tables S11–13 and Figures S20–21). The analysis indicates that $\text{NP}_B(\text{Cyt}c)$ and pep A significantly lowers the apoptosis level in comparison to the untreated samples at each dosage (p -value < 0.05). NINP, on the other hand, exhibits no significant difference from the untreated sample (p -value = 0.88 and 0.52 for the STS- and 5FU-treated cells, respectively).

- As suggested, we performed another control experiment, using NP_B(lysozyme) and found no protective effect in the apoptosis experiments. The data is added to the Supporting Information (Figure S19) and briefly discussed in the m

- Are the +/- values indicated in the ITC data tables representing standard deviations? Please clarify.

- The methods are appropriately detailed but there seem to be a lack of repeats for a number of biological experiments presented. For the ITC experiments, it is noted in the caption that the experiments have been run in triplicates. The authors should clarify whether this means 3 independent experiments, or one experiment run in triplicate. If the latter, repeats experiments need to be conducted to be able to show any S.D or SEM values.

- We included the following clarifications in the revised manuscript in the caption of Tables 1 and 2.
- Binding constants were determined by triplicate, independent ITC titrations at 298K in 10 mM HEPES buffer (pH=7.5), with the relative errors among the runs in the range of 4–7% (typically <5%). The errors shown in the Table are from curve-fitting for representative titrations, with the corresponding titration curves given in the Supporting Information.

- In other data sets, there are no SD or SEM values included (e.g. Fig 1, Fig 2, Fig 3, Fig 4, S14) suggesting the experiments have been run only once. For robustness, these experiments should be also repeated.

- Independent experiments were performed in triplicates for our data. Data in Figures 1a and 1b are represented by 3D columns and thus do not have error bars on them. In these cases, we added the following statement to the Tables 1 and 2 captions: Binding constants were determined by triplicate, independent ITC titrations at 298K in 10mM HEPES buffer (pH=7.5), with the relative errors among the runs in the range of 4–7% (typically <5%). The errors shown in the Table are from curve-fitting for representative titrations, with the corresponding titration curves given in the Supporting Information.
- For other figures, we added the errors from the independent experiments.

- Fig 5g and 5i: what are the error bars representing? Please indicate whether this data arises from independent repeats.

- We added the following statement to the figure caption: The cell experiments were repeated with three independent biological samples.

- Figure 6b and c, Fig S8, S9: here again, indicate what the error bars are representing and whether these experiments have been run several times (in independent runs). The error bars are very small for a biological experiment in cells.

- We added the following statement to the figure caption for clarification: The apoptosis experiments were repeated with three independent biological samples. The error bars in Figure 6b,c represent measurement errors in a single experiment. Statistical analysis for the biological replicates is performed using a linear mixed model and reported in the Supporting Information (Tables S11–13 and Figures S20–21). The analysis indicates that NP_B(Cytc) and pep A significantly lowers the apoptosis level in comparison to the untreated samples at each dosage (p -value < 0.05). NINP, on the other hand, exhibits no significant difference from the untreated sample (p -value = 0.88 and 0.52 for the STS- and 5FU-treated cells, respectively).

Minor comments:

Some references are little old and do not integrate the latest thinking around PPIs (e.g. references 7-9).

- We update the references for the PPIs.

P9, second last paragraph "...can strongly impact the electron transfer...": the concentrations used to achieve these effects are quite high so I wouldn't characterise this effect as "strong".

- The word "strongly" is removed in the revision.

The word "significant" is used several times across the manuscript. Yet, there is no statistical analysis to determine whether the difference between results is actually statistically significant.

- As mentioned above, in the revision, we did a full statistical analysis on the apoptosis data for all the biological replicates and included it in the Supporting Information.

The word concentration is misspelled in most of the graphs included in the supplementary info section.

- We apologize for the typos and have corrected them.

Reviewers' Comments:

Reviewer #1:

Remarks to the Author:

In the revised version, the authors modified the manuscript according to some of the comments. Unfortunately, quite a few of the major concerns raised by the reviewers have not been well addressed.

Major comments:

- 1) The authors claimed extremely high imprinting factor (IF) values, but the measurement was rather questionable. They defined the IF as the ratio of the K_a value for MIP over that for NIP. Even for strong binding for MIPs, which is more precise, the SD values for the K_a measurement could be 1.4-7.6 for MIPs (Tables 1 and 2). Since the binding for NIP is very weak, ITC is clearly not an appropriate analytical tool for its K_a value measurement. The very poor ITC titration curves for NIP shown in Figures S9 and S10 indicate that ITC fails to provide an accurate and precise K_a measurement for NIP. If one wants to make a rough estimation, then he or she must take possible SD values into account, which should be much higher than those for MIPs. As such, the actual IF could be much lower, tens and even less than 10. For protein imprinting, the IF values are usually rather hard to over 10, not to mention over 100. On the other hand, the NIPs used for the calculation of IF values must be prepared under otherwise identical conditions as MIPs (except that there is no template, other conditions should be the same). However, the NIP used in Table 2 (entry 8) contains no FM 7 at all, making the comparison much unreasonable. Overall, the characterization of the IF value and the measurement of K_a values for weak binding are unreasonable, and thereby the claim of high specificity is doubtful. It is highly suggested to evaluate the imprinting effect using binding isotherms with measurable responses, and to evaluate the cross-reactivities via measurable responses.
- 2) The mechanism for the imprinting relies on the so-called signature lysine of template protein. However, for a range of template proteins with largely varied active lysine number (6-32), the K_a values did not vary too much (Figure 1a and 2a). The K_a values for NP_a and NP_b for amylase can even as good as those for lysozyme and cytochrome C. This means that the proposed mechanism or hypothesis is not true.
- 3) The template molecules much varied in their molecular weight (12-66 kDa), so did their molecular size. However, the AM/MBAm layer in NP_b just added additional 5-6 nm to the diameter as compared with the size of NP_a. It is rather difficult to believe more or less the same layer thickness worked the same well for the templates.
- 4) Page 2, "Although direct imprinting of whole protein is highly desirable, ...". In fact, the direct imprinting of whole proteins has apparent disadvantages. The imprinting of their epitope or structural domain is more desirable, but more challenging.
- 5) The effect of surfactants on the conformational changes of template protein was only demonstrated with one protein (lysozyme). How about other template proteins?
- 6) Literature citation seems not sufficient. The binding strengths seem not to be a merit of the present approach over existing methods, because K_d values of nanomolar level even lower have been widely reported. However, if well experimentally supported, high specificity is likely an important advantage. In fact, many efforts have been made to improve the specificity of MIPs. Therefore, comparison with highly specific MIPs is highly necessary.

Reviewer #2:

Remarks to the Author:

The authors have considerably improved the manuscript in responding to the reviewer's comments. On re-reading, I am struck again by the novelty of the concept and I judge the manuscript is now appropriate for publication in Nature Communications. There are a small number of outstanding minor issues that should be addressed prior to publication.

1. Presentation of the concept and structural details in Scheme 1 is much improved. I encourage the authors to further enlarge the molecular structure diagrams for 1a, 2a, 3, and 4-8 (in the central box) as these are still quite hard to read.

2. p3, line 5. "figure 1a" should read "Scheme 1a"

3. p4, paragraph 1. The mechanistic description is much improved. However, the language in this paragraph has a misleading "active" tone. The presence of protein template does not trigger diffusion as suggested by the phrase "Our hypothesis is that, in the presence of a protein template, 1a will diffuse out of the SCM to interact with ...". The diffusion of 1a occurs without knowledge of the presence of the protein. By interacting with any 1a that happens to diffuse to the SCM-protein interface, the presence of template might bias diffusion through a Le Chatelier effect.

Likewise, the suggestion that hydrophobic voids "pull" the template into the SCM is misleading. This is not an active force, but again a characteristic that defines the energy of the internalised protein-conjugated FMs. The process of diffusion occurs without knowledge of the environment that will be encountered.

4. p7, line 10. Insert cross reference to Scheme 1b in sentence "To our delight, ..."

5. p7, line 11. There appears to be a typographical error. The first binding affinity quoted (11.8×10^5) is that achieved in absence of 7.

6. p18, line 9. The phrase "display hundreds of nanomolecular binding affinities" still requires adjusting. I don't understand what is meant by a "nanomolecular" binding affinity. I suggest the authors mean "display **dissociation constants** in the hundreds of **nanomolar** range"

Reviewer #3:

Remarks to the Author:

The authors have nicely clarified the majority of my concerns.

I am still unsure about the direct practical utility of this imprinting method although, conceptually, I find the data and results quite exciting in this field with lots of potential. The authors mentioned that the nanoparticles will be used in mechanistic studies, which is a fair point. Yet, this doesn't alleviate the need for potency and specificity, especially when used in a whole cell context.

However, as a first demonstration and proof-of-principle work, I believe this elegant paper should be published.

We are pleased that reviewers 2 and 3 consider our revised manuscript acceptable for publication after some minor issues are taken care of. Reviewer 1 raised several points which we are addressing as follows.

In response to reviewer 1:

In the revised version, the authors modified the manuscript according to some of the comments. Unfortunately, quite a few of the major concerns raised by the reviewers have not been well addressed.

Major comments:

1) The authors claimed extremely high imprinting factor (IF) values, but the measurement was rather questionable. They defined the IF as the ratio of the K_a value for MIP over that for NIP. Even for strong binding for MIPs, which is more precise, the SD values for the K_a measurement could be 1.4-7.6 for MIPs (Tables 1 and 2). Since the binding for NIP is very weak, ITC is clearly not an appropriate analytical tool for its K_a value measurement. The very poor ITC titration curves for NIP shown in Figures S9 and S10 indicate that ITC fails to provide an accurate and precise K_a measurement for NIP. If one wants to make a rough estimation, then he or she must take possible SD values into account, which should be much higher than those for MIPs. As such, the actual IF could be much lower, tens and even less than 10. For protein imprinting, the IF values are usually rather hard to over 10, not to mention over 100. On the other hand, the NIPs used for the calculation of IF values must be prepared under otherwise identical conditions as MIPs (except that there is no template, other conditions should be the same). However, the NIP used in Table 2 (entry 8) contains no FM 7 at all, making the comparison much unreasonable. Overall, the characterization of the IF value and the measurement of K_a values for weak binding are unreasonable, and thereby the claim of high specificity is doubtful. It is highly suggested to evaluate the imprinting effect using binding isotherms with measurable responses, and to evaluate the cross-reactivities via measurable responses.

- Right now, there is agreement that the binding of the proteins by the nonimprinted nanoparticles is very weak, as shown in the comment above. The disagreement is in the calculation of the imprinting factor, i.e., the binding constant of the imprinted nanoparticle for its protein template divided by the value of the nonimprinted nanoparticle (NIPs mentioned above or NINPs in the manuscript). Due to the weak binding by the NINPs, the ITC curves fit poorly and the binding constants obtained have large uncertainties. We used the ITC-estimated numbers as the upper limits for the NINP binding constants because something like 2000 M^{-1} are typically measurable and yet our NINP bindings were not.
- It is difficult to get an accurate measurement of the binding constant for any weak binding event unless one of the binding partners is used at a high concentration to shift the equilibrium to the bound state. Because proteins aggregate easily at high concentrations, we couldn't perform titrations with a high concentration of protein. In this round of revision, we attempted to titrate the protein guests with varying concentrations of NINPs but still obtained noisy titration curves consistent with nondetectable binding.
- In response to the comment, to avoid using unmeasurable values for the nonimprinted nanoparticles to calculate the imprinting factors, we removed all the mentioning of imprinting factors in the manuscript. In addition, in all the figure and table captions, whenever the binding was too weak to be measured, we added a statement so that readers are aware of the uncertainties—"Because of the weak binding, the titration curve fits poorly and has large uncertainties in the estimated binding constant." It should be pointed out that large uncertainties in a poorly fitted curve fitting do not mean the binding may be strong, as strong bindings can be measured easily. Instead, they just mean that the binding is weak and it is difficult to know the binding constant with certainty.
- We believe our revision is reasonable because, in a chemical or biological application of these protein-binding nanoparticles, the imprinting factors reflect the extent of nonspecific interactions displayed by the materials relative to the imprinted materials. Essentially, when the background noise is high, it is difficult to observe the real effect of the imprinted material, which is problematic.

In all of our *in vitro* and *in cellulo* experiments, however, the NINP controls display negligible activities (Figures 3, 4, and 6), indicating they are incapable of having the specific effects derived from the protein-imprinted nanoparticles.

- Lastly, we do have two control nanoparticles in the case of **NP_B**—the nonimprinted nanoparticles with and with the FMs. Neither showed measurable binding. In the revision, we included both of them in Table 2 (entries 8 and 9).

2) The mechanism for the imprinting relies on the so-called signature lysine of template protein. However, for a range of template proteins with largely varied active lysine number (6-32), the K_a values did not vary too much (Figure 1a and 2a). The K_a values for NP_A and NP_B for amylase can even as good as those for lysozyme and cytochrome C. This means that the proposed mechanism or hypothesis is not true.

- The reversible covalent binding between trifluoromethyl ketone and primary amine is a well-established strategy in molecular recognition and sensing (ref 41–43 and references therein). Table 1 shows that **NP_A**(lysozyme) prepared with FM **1a** binds lysozyme 62 times more strongly than the nanoparticle prepared with lysozyme but without any trifluoromethyl ketone FM. In addition, the binding constant responds to the structure of different trifluoromethyl ketone FMs, consistent with its importance.
- To further support the involvement of this binding interaction in the protein recognition, we performed ITC titrations of **NP_A**(lysozyme) by lysozyme in the presence of 0, 1, 2.5, and 5 equivalents of 3-amino-1-propanol that competes with the protein for the trifluoromethyl ketone binding groups. The binding between the protein and the NP receptor decreased from $57.3 \times 10^4 \text{ M}^{-1}$ (Table 1, entry 1) quickly to $5.0 \times 10^4 \text{ M}^{-1}$ (Figure S12a), to $2.4 \times 10^4 \text{ M}^{-1}$ (Figure S12b), and then to $<3 \times 10^3 \text{ M}^{-1}$, respectively (Figure S12c). These inhibitory experiments provide unequivocal evidence that the amine–trifluoromethyl ketone binding interaction is the key driver for the recognition. The new data is briefly discussed in the revised manuscript.
- We have offered a rationale for the largely similar binding constants between various proteins and their nanoparticle receptors in the previous version of the manuscript (see the violet colored text below). In short, they are derived from the dominant contribution of the reversible covalent bonds (hemiaminal) to the binding and the limited, similar number of such bonds that can be formed between a protein and a spherical nanoparticle. In the revised manuscript, we added a reference to the inhibitory experiments that support the importance of the covalent bonds to the binding.

As discussed earlier, our hypothesis is that dynamic covalent chemistry will favor a region of protein with multiple reactive lysines close by, among other factors. Since a spherical SCM can only accommodate and stabilize a limited number of protein-conjugated FMs due to its geometrical constraint, we expect the binding affinity of **NP_A** would not differ greatly if these FMs are the main contributors to the binding (as supported by the inhibitory experiments discussed above) and a similar number of the hemiaminal bonds are formed in all protein–NP pairs. Even if a protein contains many reactive lysines, only those with their hemiaminal of **1a** inserted into the SCM, stabilized (through water exclusion), and captured covalently (through polymerization) will contribute to the protein binding at the end.

The above postulation was confirmed experimentally. The number of reactive lysines on our protein templates vary from 6 to 32.⁴⁹⁻⁵³ If the N-terminal amine is counted, it will add 1 to the above number. Yet, the binding constants of the different NPs toward their templating proteins were $(54 \pm 4) \times 10^4 \text{ M}^{-1}$, across different proteins and their NP binders (Figure 1a).

On the other hand, as the number of reactive lysines on a protein template increases, a larger FM/template ratio is needed in the NP preparation. A small amount of FM will be unable to shift the equilibrium to the state with multiple adjacent lysines anchored into the SCM by the hemiaminals formed. This was also confirmed experimentally. As shown in Figure 1b, the

optimal FM/template ratio in the NP_A preparation correlates linearly with the number of reactive lysines on the protein templates.

3) The template molecules much varied in their molecular weight (12–66 kDa), so did their molecular size. However, the AM/MBAm layer in NPb just added additional 5–6 nm to the diameter as compared with the size of NP_a. It is rather difficult to believe more or less the same layer thickness worked the same well for the templates.

- Our data indicate that the outer AM/MBAm layer on NP_B is a minor contributor to the binding in comparison to the covalent hemiaminal bonds. (This point is also supported indirectly by the inhibitory experiments mentioned above). NP_A (lysozyme), for example, binds lysozyme (its protein template) with a binding free energy of $-\Delta G = 7.86$ kcal/mol (Table 1, entry 1). Adding the AM/MBAm layer (without the FM 7) only increases the binding constant by 2-fold, which translates to an increase of binding free energy by merely 0.41 kcal/mol. Even with the FM 7 in the AM/MBAm layer, the binding energy only increases by 1.0 kcal/mol.
- Thus, our data indicate that the major contributor to the binding is the reversible hemiaminal covalent bond and all the other interactions, whether with NP_A or NP_B , are secondary. Since the outer AM/MBAm layer only strengthens the protein binding by no more than 1 kcal/mol, molecular imprinting in the outer AM/MBAm layer is moderate with the current method, especially considering the binding interface potentially involved.
- The above discussion is added to the revised manuscript.

4) Page 2, “Although direct imprinting of whole protein is highly desirable, ...”. In fact, the direct imprinting of whole proteins has apparent disadvantages. The imprinting of their epitope or structural domain is more desirable, but more challenging.

- The reviewer’s point is well taken. In our introduction, epitope imprinting and whole protein imprinting are presented as complementary techniques, each with its own merits and weaknesses. The literature also supports the view. It is not our intention to criticize one over the other.
- As the reviewer mentioned, it can be very challenging to identify appropriate epitopes (especially for discontinuous epitopes). Under such scenarios, imprinting of the whole protein becomes highly desirable and requires no structural information on the protein.
- In the revision, we added the following sentences to make these points clear: “For discontinuous epitopes consisting of separate residues brought together by the folding of the peptide chain, it is much harder or even impossible to find a small molecule analogue that can be used as the template. Imprinting of a whole protein, on the other hand, uses the natural protein directly as the template and can yield a binder with a more extensive buried interface. No structural information is needed for the imprinting as long as the protein is available.”

5) The effect of surfactants on the conformational changes of template protein was only demonstrated with one protein (lysozyme). How about other template proteins?

- As suggested, we studied the effects of the surface-cross-linked micelles (SCMs) and imprinted nanoparticles on all the protein templates. None of the proteins displayed conformational changes according to their CD spectra (Figure S2).

6) Literature citation seems not sufficient. The binding strengths seem not to be a merit of the present approach over existing methods, because K_d values of nanomolar level even lower have been widely reported. However, if well experimentally supported, high specificity is likely an important advantage. In fact, many efforts have been made to improve the specificity of MIPs. Therefore, comparison with highly

specific MIPs is highly necessary.

- A notable challenge with traditional imprinting is the heterogenous distribution of binding sites and low percentage of high affinity binding sites. Piletsky and co-workers reported an innovative method of solid-phase synthesis of molecularly imprinted nanoparticles that allows low-affinity nanoparticles to be removed. The final nanoparticles display “more uniform binding characteristics”. This is the most notable advancement in this area in recent years. In addition, orientated immobilization of templates via boronic acids, aptamers, and other techniques has been used to improve the binding properties of protein-recognizing MIPs.
- We briefly discussed these methods in the Introduction. Molecular imprinting has a long history with many developments. If we miss any key papers that are relevant to the current study, we will appreciate it if the reviewer bring them to our attention.

In response to reviewer 2:

The authors have considerably improved the manuscript in responding to the reviewer’s comments. On re-reading, I am struck again by the novelty of the concept and I judge the manuscript is now appropriate for publication in Nature Communications. There are a small number of outstanding minor issues that should be addressed prior to publication.

- We thank the reviewer for the kind words of encouragement.

1. Presentation of the concept and structural details in Scheme 1 is much improved. I encourage the authors to further enlarge the molecular structure diagrams for 1a, 2a, 3, and 4-8 (in the central box) as these are still quite hard to read.

- As suggested, we split the central box into two so that we could increase the height of the figure relative to the width. This also allows us to expand the entire picture to the page width, which further expands everything.

2. p3, line 5. “figure 1a” should read “Scheme 1a”

- The typo was corrected.

3. p4, paragraph 1. The mechanistic description is much improved. However, the language in this paragraph has a misleading “active” tone. The presence of protein template does not trigger diffusion as suggested by the phrase “Our hypothesis is that, in the presence of a protein template, 1a will diffuse out of the SCM to interact with ...”. The diffusion of 1a occurs without knowledge of the presence of the protein. By interacting with any 1a that happens to diffuse to the SCM-protein interface, the presence of template might bias diffusion through a Le Chatelier effect.

Likewise, the suggestion that hydrophobic voids “pull” the template into the SCM is misleading. This is not an active force, but again a characteristic that defines the energy of the internalised protein-conjugated FMs. The process of diffusion occurs without knowledge of the environment that will be encountered.

- We modified the description as suggested.

4. p7, line 10. Insert cross reference to Scheme 1b in sentence “To our delight, ...”

- We added the cross reference as suggested.

5. p7, line 11. There appears to be a typographical error. The first binding affinity quoted (11.8×10^5) is that achieved in absence of 7.

- We thank the reviewer for catching the typo. The typo was corrected.

6. p18, line 9. The phrase “display hundreds of nanomolecular binding affinities” still requires adjusting. I don’t understand what is meant by a “nanomolecular” binding affinity. I suggest the authors mean “display **dissociation constants** in the hundreds of **nanomolar** range”

- Biologists frequently refer to the dissociation constant as binding affinity. We made the suggested changes so that it can be clearly understood by researchers in different disciplines.

In response to reviewer 3:

The authors have nicely clarified the majority of my concerns.

I am still unsure about the direct practical utility of this imprinting method although, conceptually, I find the data and results quite exciting in this field with lots of potential. The authors mentioned that the nanoparticles will be used in mechanistic studies, which is a fair point. Yet, this doesn't alleviate the need for potency and specificity, especially when used in a whole cell context. However, as a first demonstration and proof-of-principle work, I believe this elegant paper should be published.

- We thank the reviewer for the kind words of encouragement.

Reviewers' Comments:

Reviewer #1:

Remarks to the Author:

The authors have well improved the manuscript in responding to the reviewers' comments. Now it has been a well supported paper and can be accepted as is.